# CERTIFIED DEDUCTIVE REASONING WITH LANGUAGE MODELS

## ABSTRACT

Language models often achieve higher accuracy when reasoning step-by-step in complex tasks. However, even when arriving at a correct final answer, their rationales are often logically unsound or inconsistent. This is a major issue when reliable reasoning traces are needed, such when fine-tuning on model-generated reasoning for self-improvement. To tackle these issues, we introduce a class of tools for language models called *guides*, that use state and incremental constraints to guide generation. A guide can be invoked by the model to constrain its own generation to a set of valid statements given by the tool. In turn, the model's choices can change the guide's state. We show how a general system for logical reasoning can be used as a guide, which we call LOGICGUIDE. Given a reasoning problem in natural language, a model can formalize its assumptions for LOGICGUIDE and guarantee that its step-by-step reasoning is sound. In experiments on PrOntoQA, ProofWriter and Syllogism Validity datasets, LOGICGUIDE significantly improves the performance of GPT-3, GPT-3.5 Turbo and LLaMA (accuracy gains up to 35%), while drastically reducing *content effects* — the interference between unwanted prior assumptions and reasoning, which humans and language models suffer from. We then explore bootstrapping GPT-3.5 Turbo and LLaMA using their own reasoning traces. We find that LogicGuide is critical: by training only on certified self-generated reasoning, models can self-improve, avoiding learning from their own hallucinations. Moreover, bootstrapped models enjoy significant boosts on ReClor, a challenging real-world reasoning dataset, even when not relying on formalization at inference time.

## 1 INTRODUCTION

Consider a language-based autonomous agent tasked with managing a user's calendar and email. The user might want to specify general principles on how the agent should behave, such as "if the email is from any of my managers, you must send me a notification", and important pieces of information such as "I'm part of the research team", or "Grace manages research". When the agent analyzes an email and decides what actions to take, we'd like it to respect the given instructions. Doing so might require *reasoning*: the agent should conclude that an email from Grace warrants a notification, even if that wasn't said explicitly. How should the agent draw such conclusions?

A Large Language Model (LLM), such as GPT-3 (Brown et al., 2020) or PaLM (Chowdhery et al., 2022), can in principle take in the given instructions and context, choose actions to take and, before each action, ask itself "is this permitted?" The answer might require making chains of inferences based on the user's input. For this class of problems, LLMs have been shown to dramatically benefit from chain-of-thought reasoning (Wei et al., 2022; Suzgun et al., 2022). Empirically, allowing LLMs to generate reasoning steps before their answer consistently yields higher accuracy across a wide range of tasks (Suzgun et al., 2022). Qualitatively, reasoning steps are often seen as "an interpretable window" into how the model arrived at the answer (Wei et al., 2022), in contrast to an opaque guess.

But much like humans, language models can also produce unsound reasoning: even after correctly interpreting a problem, they can take logically invalid inference steps, or produce a guess at the final answer that is not supported by their own rationale (Saparov & He, 2022). Moreover, LLMs have also been observed to show human-like *content effects* in reasoning: their accuracy drops significantly when asked to reason with assumptions that contradict their prior beliefs (Dasgupta et al., 2022).

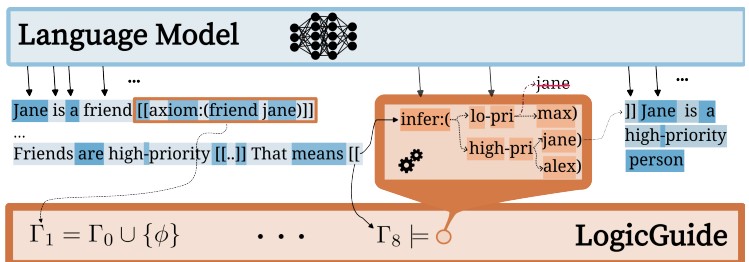

Figure 1: A language model can invoke a guide tool, such as our LOGICGUIDE, to perform certifiable generations. Here, when the model decides to generate an `infer` block, it is constrained to generate one of the formal deductions established by an external theorem-proving environment.

How can we avoid unsound, perhaps dangerous, inferences? This question illustrates the central concern that led to the development of formal logic. Proof systems formally encode core patterns underlying valid reasoning, allowing sound inferences to be generated mechanically from deduction rules. If we arrive at an answer after a series of formal deductions, we know that not only the conclusion is correct, but the *reasoning* that led to it was entirely valid. This is in contrast to a free-form rationale that can arrive at the correct answer even through incorrect means.

To that end, we aim to allow LLMs to rely on trusted formal deductions during reasoning by building on the recent paradigm of *tool use* in language models (Cobbe et al., 2021; Schick et al., 2023). In prior work, LMs invoke external tools by generating special sequences, intercepted by the decoding algorithm. They can generate inputs (e.g., a mathematical operation, or search query) and receive the tool's output as if it was their own generation. We generalize this input-output paradigm to a broader class of LM tools we call *guides*. When a guide is invoked by the model using a special delimiter, the tool computes a space of valid outputs, as illustrated in Fig. 1. We then employ *constrained decoding* (Poesia et al., 2022) to ensure the model will incrementally generate one of the valid outputs. Guides thus enable a more declarative interaction between tool and model: the guide declares a set of possible sequences, while the model brings prior expectations used to generate one among them. A guide can maintain state: its next valid outputs might depend on the sequence of choices up to that point.

We use this framework to allow language models to locally constrain generation to a set of valid statements determined by an external tool. We leverage the Peano theorem-proving environment (Poesia & Goodman, 2022) to construct LOGICGUIDE, which an LM can use to formalize its assumptions, set proof goals and make sound inferences step-by-step. The model can intersperse formal reasoning and natural language during generation. Language conditioned on previous formal steps is highly reliable, since the generations allowed by LOGICGUIDE are formally certified.

We first validate our method on three logical reasoning datasets, PrOntoQA (Saparov & He, 2022), ProofWriter (Tafjord et al., 2021), and Syllogistic Validity (Dasgupta et al., 2022). We also follow the format and methodology of PrOntoQA to introduce a new dataset, DeontiQA, where problems require reasoning using deontic logic principles to determine whether actions are permissible, obligatory or forbidden. When used with few-shot prompting, we find that LOGICGUIDE significantly improves the accuracy of OpenAI GPT-3 and GPT-3.5 Turbo, and LLaMA 13B. Moreover, models using LOGICGUIDE have drastically lower content effects: we show this both with PrOntoQA and in the Syllogism Validity dataset, used previously to measure content effects in LLMs.

While for these problems we could leverage an external solver to obtain a final answer after the model produces a sufficient formalization, a key advantage of LOGICGUIDE is that the LLM still generates a step-by-step rationale. This allows us to then apply self-improvement methods, such as the Self-Taught Reasoner (STaR; Zelikman et al. (2022)), which improves the model's own reasoning by fine-tuning on rationales that led to correct answers. In the tasks we analyze here, there's a high probability of guessing the answer (e.g. true or false, so at least 50%). Hence, STaR alone fails to yield meaningful improvements, as it fine tunes on incorrect reasoning that does not generalize. In contrast, we show that running STaR using only certified solutions generated with LOGICGUIDE is highly effective: LLaMA 13B enjoys accuracy gains of up to 17% on PrOntoQA, while naïve STaR fails to improve the model.

Finally, we investigate whether bootstrapped models learn reasoning patterns that generalize beyond problems where full formalization is possible. We fine-tune GPT-3.5 Turbo on its own correct solutions for problems in PrOntoQA and ProofWriter and evaluate it on ReClor, a challenging set of problems from real-world standardized exams requiring reading comprehension and logical reasoning, as well as 6 tasks in the LEGALBENCH dataset. Bootstrapped GPT-3.5 Turbo outperforms both the base model, and performs best when fine-tuned on its solutions generated with LOGICGUIDE. These results suggest a promising avenue for bootstrapping language models on logical reasoning.

## 2  RELATED WORK

Our work builds on two classes of systems for reasoning: language models, which can reason flexibly in natural language, and formal reasoning systems, which rely on formal logic to derived certified inferences. To interface these two systems, we leverage recent methods for constrained decoding from language models. Specifically, we employ Constrained Semantic Decoding (CSD, Poesia et al. (2022)), an algorithm that guarantees valid samples by construction. CSD does not require full access to the model, only the ability to bias its logits. This allows us to use GPT-3 and GPT-3.5 Turbo through their public APIs, as well as LLaMA (Brown et al., 2020; Touvron et al., 2023). Other decoding methods, such as NeuroLogic Decoding (Lu et al., 2021) and NeuroLogic A*esque decoding (Lu et al., 2022), have been proposed to enforce lexical constraints at inference time.

LLMs have been increasingly used as agents interacting with other systems, by both using tools to delegate computation or to trigger external actions (Schick et al., 2023; Yao et al., 2022; Yang et al., 2023; Hosseini-Asl et al., 2020; Shah et al., 2023). In prior work, LLMs can provide inputs to an external tool, such as a search query (Schick et al., 2023) or a mathematical operation (Cobbe et al., 2021), and receive the output in the decoding stream. Our framework of guides (§3) can be seen as a generalization of this paradigm, where the tool defines a space of outputs and the LM chooses one using its own probabilities.

Our approach to certifying reasoning from LMs relies on grounding their inferences in an interactive theorem prover, Peano (Poesia & Goodman, 2022). Similar to other popular theorem proving languages like Lean (de Moura et al., 2015) and Coq (Barras et al., 1997), Peano uses dependent type theory as its logical foundation. Most theorem proving environments are designed for the *verification* of given proofs. In contrast, and of special interest to us, Peano is designed to aid in *generation* by exposing a finite action space. Many other recent works have integrated LLMs and interactive theorem provers in the context of formal mathematical reasoning. Recent work on autoformalization has shown that LLMs can be effective in translating informal to formal mathematics (Wu et al., 2022). This idea is related to how we use LLMs to formalize their assumptions given in natural language, though our end goal is to produce reliable natural language rationales rather than formal proofs.

Many prior works have broadly used neural networks for logical reasoning. One prominent line of work has focused on using neural networks to approximately execute logical reasoning on symbolic knowledge bases. This includes Neural Theorem Provers (Rocktäschel & Riedel, 2017) and Conditional Theorem Provers (Minervini et al., 2020). Other approaches use encoders for allowing problem statements in natural language and perform reasoning in latent space, such as Discourse-Aware Graph Networks (Huang et al., 2021). Unlike both approaches, which learn to perform reasoning "in-weights" via fine-tuning, our goal is to augment chain-of-thought reasoning in language models, where all inputs, reasoning steps and outputs are realised in language.

## 3  CERTIFIED REASONING WITH GUIDES

Previous work in tools for language models assumed an interface where the model provides inputs to the tool and receives back a single output, conditioning on this output for further generation. For instance, Cobbe et al. (2021) allowed the model to rely on a calculator by generating a string such as «51*12=. At this point, the decoding algorithm would execute the operation externally and copy the result as if it was generated by the language model. Here, our main goal is to leverage a trusted external tool to answer the question: "what logical inferences can be made next?" Unlike an arithmetic operation, this question (1) can have a potentially large set of answers, and (2) can depend on previous choices made by the model. Thus, our key idea is to leverage *constrained generation* to

allow the model to implicitly choose one of the valid inferences during decoding, while remaining contained in those that are valid.

More generally, a guide tool defines a set of valid generations given previous choices. Formally, let $S = \Sigma^*$ be the set of strings in the guide's alphabet $\Sigma$, with $S^*$ denoting the set of finite sequences of such strings. We define a guide $g$ to be a function $g : S^* \rightarrow \mathcal{P}(S)$ that takes a sequence of previously generated strings and returns a regular set of allowed next generations. Our idea is to use $g$ at specific points when sampling from a language model $P_{LM}(\cdot)$ so that when the guide is invoked at a prefix $s_0$, we will sample a continuation from $P_{LM}(s|s_0)$ that belongs to the set allowed by $g$ when given the previous guided generations in the prefix $s_0$ (e.g., previous valid logical inferences).

## 3.1  FROM GUIDES TO COMPLETION ENGINES

Given any guide function $g$ as above, we want to provide a tool for LMs that, once invoked by a special sequence, will constrain the immediate subsequent output using $g$. Let $t_1$ and $t_2$ be two arbitrary delimiters, such that $t_1$ begins a guided block in the LM's generation, which is then closed by $t_2$ (e.g., we later use $t_1 = $ "`[[`" and $t_2 = $ "`]]`"). Intuitively, we would like to decode from $P_{LM}$ with the following idea: (1) we sample tokens until the model generates $t_1$; once that happens, we (2) use $g$ along with the generation so far to obtain a set of valid continuations, then (3) trigger constrained generation to sample from that set, and finally (4) return to unconstrained generation once the model outputs $t_2$. Unfortunately, tokenizers complicate implementing this procedure, as different models often have different vocabularies (like all 3 models we leverage in §4 do), and LM tokens can be arbitrarily misaligned with $t_1$ and $t_2$. For example, the string "`[[`" might be a single token, contained in a larger token, or be broken up across two tokens depending on the context.

To overcome these issues and implement LM guides in a model-agnostic manner, we employ the Constrained Semantic Decoding algorithm (CSD; Poesia et al. (2022)). CSD effectively solves the vocabulary alignment problem by providing the abstraction of a "completion engine", which incrementally dictates valid generations by constructing local regular expressions. CSD samples from the model while guaranteeing a valid output (i.e., one that respects the completion engine). We formally define how we construct a completion engine to implement guide tools in the Appendix. Using our implementation, however, users can simply implement an arbitrary function $g$ with the signature above and obtain a procedure to sample from any LM with the guide being invoked when needed. This framework allows us to easily design rich, context-sensitive LM tools, as the LOGICGUIDE we introduce next. We describe several other guides in the Appendix, leaving their exploration for future work.

## 3.2  THE LOGICGUIDE

We now construct LOGICGUIDE, a guide tool for language models to perform externally certified reasoning. Our logical backend of choice is Peano (Poesia & Goodman, 2022), a theorem-proving environment for incremental proof generation. The main feature of Peano we rely on is that it provides a finite action space. Thus, given a partial argument, Peano gives us a list of valid inferences that can be made in a single step given the background theory, assumptions (possibly previously added by the model) and past inferences. Our idea is to use this list to guide the model whenever it decides to derive a logical conclusion. While Peano makes the guide implementation particularly simple, other theorem-proving environments might be adaptable for this purpose.

We use the delimiters "`[[`" and "`]]`", and implement a guide function that accepts strings with the format `action:parameter`. We define 6 actions (exemplified in Fig. 2) that allow the model to (1) formalize its assumptions (`object`, `prop`, `relation`, `axiom`), (2) set a goal (`goal`), and (3) perform logical inferences (`infer`). For (1) and (2), the guide returns constraints that ensure the model's formalization to be *syntactically* valid. Since these actions are the boundary between natural and formal language, it is impossible to guarantee that they are *semantically* valid in the sense that the model has properly interpreted the natural language context. What *is* certifiable is that the logical inferences (action type 3) follow from the model's formalization, i.e. its inference are valid given its explicit assumptions. (§4 provides empirical evidence that formalization errors rarely lead to a wrong conclusion; most often they make the model unable to prove or disprove the goal).

> **Context:** 1- The mouse visits the tiger. (...) 3- If something visits the tiger then it visits the mouse. (...) 12- If something visits the mouse then it is blue. (...)
> **Question:** True or false: The mouse is green?
>
> **Formalized context:** 1- The `[[object: mouse]]` `[[relation: visits]]` the `[[object: tiger]]` `[[axiom: (visits mouse tiger)]]` (...) 3- If something `[[relation: visits]]` the `[[object: tiger]]` then it `[[relation: visits]]` the `[[object: mouse]]` `[[axiom: (visits 'x tiger) -> (visits 'x mouse)]]` (...) 12- If something `[[relation: visits]]` the `[[object: mouse]]` then it is `[[prop: blue]]` `[[axiom: (visits 'x mouse) -> (blue 'x)]]` (...)
>
> **Formalized goal:** `[[goal: (green mouse)]]`
>
> **Reasoning:** `[[infer: (visits mouse mouse)]]` The mouse visits itself. `[[infer: (blue mouse)]]` The mouse is blue. `[[infer: (green mouse)]]` The mouse is green. This satisfies the goal.
>
> **Answer:** TRUE

Figure 2: Example solution of `gpt3.5-turbo` using LOGICGUIDE in a problem from ProofWriter. The model's generation starts at the "Formalized context". This example shows all 6 actions we implement in the LOGICGUIDE: `object` declares a particular entity, `prop` and `relation` mark unary and binary predicates, respectively; `axiom` denotes propositions assumed to hold (possibly implications), `goal` sets a target to be proven or contradicted, and `infer` marks deductions.

**Using the guide** In the typical setup for logical reasoning problems (Tafjord et al., 2021; Saparov & He, 2022), the input contains a context (the set of assumptions) and a question (a goal that the logical argument must conclude or negate). In our experiments, we demonstrate how to use the guide by creating few-shot examples with the proper LOGICGUIDE action annotations (as in Fig. 2). Specifically, we add a section before the rationale named "Formalized context" where we repeat the assumptions in the scenario while marking objects, properties and relations, and formalizing each of the assumptions into an `[[axiom:]]` block. We do the same for the goal. Then, we prepend each reasoning step with the appropriate `[[infer:]]` action. In this way the model is encouraged to first generate a formal inference step and only then its natural language counterpart. We include all of our prompts in the Appendix.

## 4 EXPERIMENTAL EVALUATION

We now evaluate the effectiveness of LOGICGUIDE in improving language models on reasoning tasks. (Our code and data are available at `github.com/<<redacted>>`). We focus on four research questions: **(RQ1)** Does LOGICGUIDE improve the accuracy of language models in multi-step reasoning? **(RQ2)** Does LOGICGUIDE reduce content effects in language model reasoning? **(RQ3)** Can an LLM self-improve using LOGICGUIDE by learning from its own solutions? **(RQ4)** Do bootstrapped models also improve in tasks where they cannot rely on LOGICGUIDE during inference?

### 4.1 IMPACT OF LOGICGUIDE IN MULTI-STEP REASONING ACCURACY

**Datasets** We first use two recent natural language reasoning datasets: PrOntoQA (Saparov & He, 2022) and ProofWriter (Tafjord et al., 2021). Both datasets contain reasoning problems with (1) a list of assumptions (e.g. "Every dog is a mammal", or "Sam is a dog"), and (2) a proposition that can be reasoned about from the assumptions (e.g. "Sam is a mammal?"). In both datasets, the goal is to answer the question with either true or false. Problems are categorized by how many reasoning "hops" the solution needs (1 to 5). In addition, PrOntoQA has three splits: "True Ontology", where the rules are coherent with common sense, "False Ontology", where rules violate commonsense (e.g., "Every composite number is prime"), and "Fictitional Ontology", which uses made-up concepts (e.g., "Every wumpus is feisty."). ProofWriter uses real concepts for all rules (e.g., people, animals, colors), but the rules are generated at random—thus they also often contradict commonsense. We use the problems from ProofWriter where the answer can be proved (i.e. ignoring the "closed-world assumption" and "unknown" problems, where fully justifying the answer requires meta-logical reasoning).

**Language models** We evaluate three language models in the few-shot setting: OpenAI GPT-3 (`text-davinci-003`; Brown et al. (2020)), OpenAI GPT-3.5 Turbo (`gpt-3.5-turbo`) and LLaMA 13B (Touvron et al., 2023). We use 4 few-shot examples for the vanilla models. For guided models, the prompt examples are augmented to show formalized reasoning. In the prompt, we first show the model how to formalize the assumptions and the goal, and then present the chain-of-thought

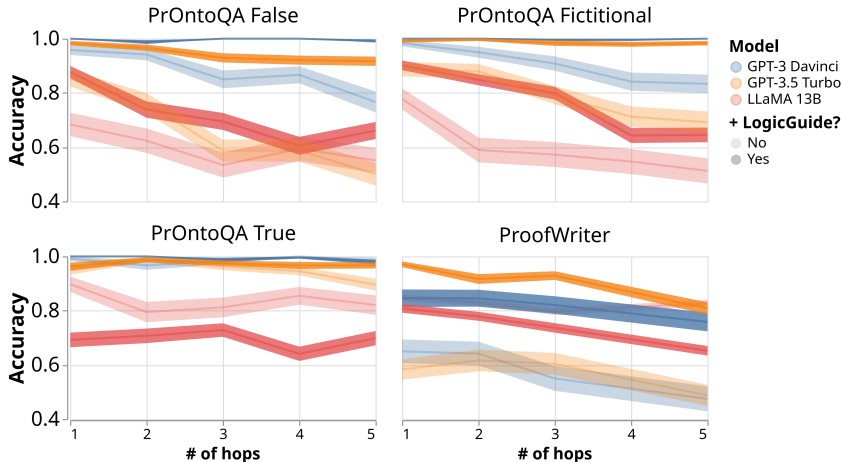

Figure 3: Final answer accuracies with guided and unguided language models on PrOntoQA and ProofWriter, with bootstrapped 95% confidence intervals.

where natural language sentences are preceded by a guided inference (in an `infer` block, c.f. §3.2). Since this makes the prompt longer, we only use two prompt examples for the guided models: one where the answer is true and one where it is false. We implement CSD on the OpenAI models using their public API, which exposes a parameter to bias the logits on given tokens. We use the rejection-based sampling procedure described in Poesia et al. (2022). `gpt3.5-turbo` requires a slight adaptation (to resume generation after a constraint is applied) because of its chat-based API; we detail this along with all of our prompts in the Appendix.

**Results** Fig. 3 shows few-shot results on multi-hop reasoning, measuring final-answer accuracy. Overall, guided models perform significantly better. GPT-3 and GPT-3.5 are highly accurate in formalizing assumptions, and enjoy the largest benefits (with nearly perfect performance on PrOntoQA with LOGICGUIDE, and improving from chance to 80% correct on ProofWriter). For them, LOGICGUIDE essentially eliminates single-step reasoning errors, and the impact of this benefit grows in solutions requiring more hops—a single error is enough to reach the wrong final conclusion. LLaMA 13B sees gains between 10 and 20% in PrOntoQA False and Fictitional, while LOGICGUIDE hurts its performance in PrOntoQA True (where, effectively, reasoning is not necessary, only commonsense) and ProofWriter (where LLaMA is more often inconsistent in its formalization).

We observe two main failure modes: (1) models can misformalize assumptions, and (2) they can fail at *planning*, making a sequence of valid inferences that do not ultimately lead to the goal. When formalization errors happen, it's more common that no conclusion can be drawn, rather than a wrong conclusion: in only 1.6% of the solutions did a guided model formally derive a wrong answer; these cases were mostly due to missing a negation when formalizing a sentence (mostly LLaMA on ProofWriter). A more common formalization failure (especially for LLaMA) was to use inconsistent names for properties or relations, e.g. `(sees A B)` in one place and `(see B C)` in another. When no further inferences can be made, LOGICGUIDE generates the string `nothing` in the `[[infer]]` block. When that happens, we observed models spontaneously concluding that the answer is "Unknown" or "Cannot be concluded" *despite that not being demonstrated in the prompt* (models abstained in 78% of the cases where they exhausted the inferences that could be made). This contrasts with the unguided models, which most often still make an unjustified guess, writing as if it was a logical conclusion (only unguided GPT-3.5 Turbo ever abstained, in 9% of its predictions).

**DeontiQA** Errors in LLM reasoning would be especially problematic when an agent must decide which actions are allowed by its instructions. Hence we created DeontiQA: a set of 60 new reasoning problems inspired by Deontic Logic (Von Wright, 1951). Deontic Logic is concerned with judgements of the type "action X is permissible/obligatory" (or not), rather than solely "proposition X is true" (e.g., in first-order logic). Peano allows us to easily embed the deontic axioms on top of its type theory. We follow the methodology used in PrOntoQA to create the problems, creating logical forms

Figure 4: Accuracies of models with and without LOGICGUIDE on the Syllogism Validity task.

first and then realizing them in natural language. Like in PrOntoQA, we add distractor rules to prevent guessing the answer from surface shortcuts. In these problems, the goal is to decide whether a given action is permissible, obligatory, or impermissible in the context of managing calendar events for a group of people. We detail the creation of DeontiQA in the Appendix, and make the dataset available along with our code. DeontiQA problems are significantly longer (up to 28 rules) compared to PrOntoQA (maximum of 18). This increased length means we are only able to fit one prompt example in the context window of GPT-3 and GPT-3.5 Turbo. We find LOGICGUIDE to be helpful on DeontiQA: GPT-3 alone is correct on 61.6% of problems, which increases to 80.0% with LOGICGUIDE. GPT-3.5 Turbo achieves 66.7% accuracy which increases to 78.3% when guided.

Overall, this provides positive evidence for our first research question: LOGICGUIDE *can significantly improve the accuracy of base models in natural language reasoning problems*. Their answers become not only more accurate but also more trustworthy: LOGICGUIDE makes models answer "Unknown" when they don't have an answer, rather than producing an unsupported guess.

## 4.2 MITIGATING CONTENT EFFECTS IN REASONING

Both humans (Evans, 2002) and language models (Dasgupta et al., 2022) have been shown to suffer from *content effects* in reasoning: their accuracy in logical judgements is influenced by prior beliefs about the assumptions and conclusions. For instance, from the assumptions that "Some librarians are happy people" and "Some happy people are healthy people", it does not logically follow that "Some librarians are healthy people". Humans and LMs have difficulty judging this argument as invalid because the conclusion agrees with prior beliefs. We hypothesize that LMs will have smaller influence from the content when *formalizing* assumptions, rather than *reasoning* from logical sentences. If that is the case, then using LOGICGUIDE will help mitigate content effects.

We use two tasks to investigate this hypothesis. First, we contrast the results in the different PrOntoQA ontologies. As in the original PrOntoQA results (Saparov & He, 2022), we see that the base performance of GPT-3 and GPT-3.5 Turbo is already close to ceiling in the True Ontology split (where the model doesn't strictly need to reason correctly as long as it judges the conclusion using common sense). In contrast, accuracy is significantly lower in the False and Fictitious ontologies and decays with more hops. However, both of these models are highly accurate in formalizing assumptions, and thus benefit from the guide in the False and Fictitious ontologies: performance is near ceiling. Interestingly, GPT-3.5 Turbo still exhibits occasional content effects, explicitly judging the conclusions derived using LOGICGUIDE as nonsensical in cases where they do follow from the problem. For instance, in one problem where the model must decide whether Sam is luminous or not, it is given that "Sam is a snake", and from the given assumptions the model correctly concludes "... [[infer:(sheep sam)]] Sam is a sheep". It then proceeds to question this conclusion and halts: "This contradicts the fact that Sam is a snake. Therefore, we cannot infer whether Sam is luminous or not.".

Second, we leverage the Syllogism Validity dataset (Dasgupta et al., 2022). In this task, the model is given two assumptions and a conclusion, and has to decide if together they constitute a valid argument (i.e., the conclusion logically follows from the assumptions). The example above about librarians is taken from this dataset. Solutions have a single step: judging the argument as valid or invalid. When using LOGICGUIDE, we prompt the model to first perform a single inference given its formalization of the assumptions and then judge the validity of the argument. Syllogism Validity has 3 conditions: "Nonsense", where rules are about made-up concepts, "Consistent", where the conclusions agree with commonsense regardless of whether the argument is valid, and "Inconsistent", where the conclusion always violates world knowledge. Unguided models behave consistently with those in Dasgupta et al. (2022): in the "Consistent" split, all models strongly tend to judge the argument as being

valid, thus performing close to chance (GPT-3.5 Turbo is slightly better, at 60%). Both GPT-3 and GPT-3.5 Turbo are, however, highly accurate at formalizing the assumptions and conclusions and tend to trust LOGICGUIDE, nearing ceiling performance for all conditions. LLaMA 13B has much more difficulty judging the syllogisms, performing near chance in all conditions. However, it is still successful at formalizing many syllogisms, obtaining non-trivial performance (60% to 77%) when using LOGICGUIDE. In failure cases, it often confuses logical connectives (e.g., formalizing "Some X are Y" as "X implies Y" and vice-versa). We overall see positive evidence for our second research question: models with LOGICGUIDE show greatly diminished content effects, with stronger benefits for models that are more capable of interpreting individual sentences (despite struggling to reason with them when unguided).

## 4.3 LEARNING TO REASON BY GUIDED SELF-IMPROVEMENT

We consider improving the reasoning ability of the base language model. When using LOGICGUIDE, the model still produces its rationales, though with constraints (in contrast to approaches that aim to fully offload reasoning to an external tool). Thus, this lets us use successful rationales to improve the model itself. This is the essence of the Self-Taught Reasoner (STaR; Zelikman et al. (2022)) a simple method for improving LLM reasoning that has been shown to be effective in symbolic, mathematical and commonsense reasoning. Given a set of problems paired with correct final answers (but not reasoning traces), STaR iterates between (1) solving problems with chain-of-thought prompting, and (2) fine-tuning the model on its own generated rationales that led to correct final answers. This allows the model to improve its reasoning from a small seed set of few-shot examples.

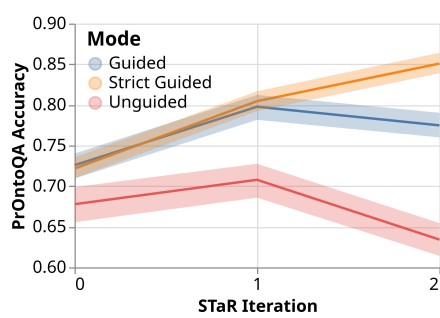

Figure 5: Accuracy of LLaMA 13B on held-out PrOntoQA problems when bootstrapping using STaR.

Crucially, STaR relies on the premise that *if a rationale led to the correct answer, it is likely to be correct*. While this holds in domains like arithmetic, it breaks down in most logical reasoning tasks. In these cases, right answers will happen often with bad rationales, leading STaR and similar approaches to fine-tune on incorrect reasoning that generalizes poorly. Indeed, the authors in Zelikman et al. (2022) remark that "filtering bad reasoning paired with correct answers remains an open question."

We thus consider STaR training on either all correct answers (with LOGICGUIDE or not) or only on certified correct answers. We run 2 STaR iterations with LLaMA 13B on PrOntoQA[1], where we attempt 200 random problems equally split between 1 and 5 hops, and fine-tune on successful solutions, evaluating on unseen problems.

Fig. 5 shows the results. As predicted in Zelikman et al. (2022), the high chance of guessing confounds STaR, and training on all rationales that yield the right answer does not give meaningful improvements ("Unguided", red curve). Training on all guided solutions leading to correct answers brings improvements ("Guided"; 72% to 80% after one iteration), but still ends up hitting accidentally-correct reasoning, when the model decides to make a guess after reasoning for a few steps. Fine-tuning only on certified correct answers avoids this trap and achieves high performance ("Strict Guided", up to 86%). This allows us to positively answer our third research question: LOGICGUIDE *can be used for effective self-improvement in reasoning*, in cases where naïve methods collapse.

## 4.4 GENERALIZING BEYOND FORMALIZATION

Finally, we consider whether models bootstrapped on their own reasoning in synthetic multi-step reasoning tasks can generalize to settings requiring similar reasoning with real-world language. To that end, we consider ReClor (Yu et al., 2020), a dataset of logical reasoning problems taken from standardized exams (such as LSAT and GMAT, 4-way multiple choice), as well as the 6 tasks in

---

[1]ProofWriter has shortcuts that allow guessing the answer without reasoning (Zhang et al., 2022), which fine-tuning quickly learns. PrOntoQA avoids those with distractor rules. Thus, we focus on PrOntoQA here.

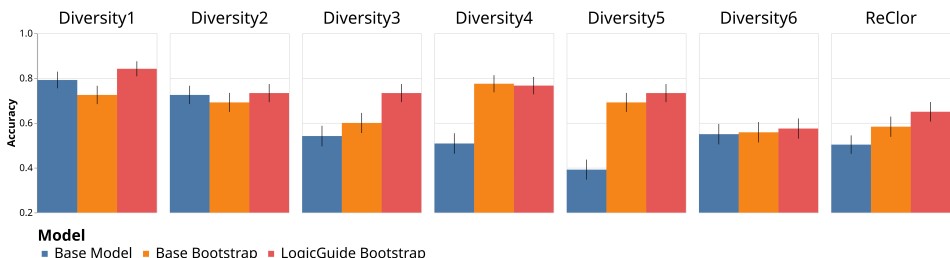

Figure 6: Zero-shot performance on six LEGALBENCH tasks and ReClor, compraing the base gpt-3.5-turbo model and as its versions bootstrapped with and without LOGICGUIDE.

LEGALBENCH (Guha et al., 2023) related to Diversity Jurisdiction (binary choice - given facts about plaintiffs, defendants and claims, determine whether the criteria for diversity jurisdiction are met). These questions are challenging even for humans. Although the questions require logical thinking, it is often unclear how to formalize them. Thus, directly using LOGICGUIDE with few-shot prompting is of limited use, and bootstrapping directly on ReClor unlikely to get off the ground. We thus explore whether models bootstrapped from formalizeable tasks will transfer to ReClor and LEGALBENCH.

Since these questions are very rich in terms of reading comprehension, we need a strong base model with non-trivial zero-shot performance. Thus, we use GPT-3.5 Turbo, and leverage the OpenAI fine-tuning API. For bootstrapping, we use a random sample of 120 correct solutions from a mixture of ProofWriter and PrOntoQA problems with 3+ hops, where the original model either used LOGICGUIDE or not. For inference we use zero-shot prompting, where we ask the model to first carefully analyze each of the options before deciding which is correct (prompt in the Appendix). Fig. 6 shows the results. Bootstrapping on solutions obtained with LOGICGUIDE yields the highest accuracy on ReClor (65%), compared to bootstrapping without LOGICGUIDE (58.3%) and to the base model (52.5%). The overall relative performance between the models is similar on LEGALBENCH tasks. When inspecting the solutions, we find that the model never explicitly tried to use LOGICGUIDE (which we would see with bracket annotations) but the style of reasoning resembles the solutions to the synthetic problems when appropriate (e.g., "Reasoning: Liam and Amelia are from different states. The amount-in-controversy is greater than $75k. Answer: A. Yes, there is diversity jurisdiction.", whereas for this same problem the base model outputs a 132-word long rationale). This style is similar in both bootstrapped models (we show several complete samples for ReClor in App. G). The higher-quality rationales obtained with LOGICGUIDE seem to have a better overall effect, leading to higher accuracy. Overall, we find positive evidence for our last research question: bootstrapping models with LOGICGUIDE can lead to better performance even when LOGICGUIDE is not available at inference time.

## 5 DISCUSSION AND CONCLUSION

We introduced guide tools, which locally constrains generation when invoked by a language model. LOGICGUIDE leveraged this idea for logical reasoning, allowing the LM to formalize its interpretation of input sentences and make sound inferences. Moreover, when bootstrapping models on their own reasoning, they can generate better reasoning even when unguided at inference time.

The direct application of LOGICGUIDE is challenging for general natural language, which is often ambiguous and can be difficult to faithfully represent in formal logic. Domains where arguments tend to have more systematic logical structure, such as law, are more likely to benefit from tools like LOGICGUIDE. Still, our work suggests that bootstrapping on formal problems might help models generalize. Even then, models can still fail at planning even when making correct deductions. Many current investigations into planning techniques for LM reasoning are complementary to our work and can be integrated with guides (Mehran Kazemi et al., 2022; Zhao et al., 2023).

Language models bring to reasoning the flexibility of human language and a wealth of useful prior knowledge. But that power comes with lack of reliability and difficulty verifying extended reasoning. Our approach points to a rich direction for seamlessly integrating reliable symbolic and flexible neural reasoning into a unified text stream. The result is better, and more easily verified, reasoning.

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

## A    OTHER GUIDES

In §3.2 we explored LOGICGUIDE, which captured a rich set of operations that both set and leverage state, as well as a complex external logical support tool. Nonetheless, many other guides can be easily defined in this same framework. We survey several such guides here as potential ideas for future work:

**Memory Guide**  A simple guide in the same format of LOGICGUIDE can be one where the model can set values to keys (`[[set:key=value]]`), and later on retrieve the value associated with a given key (`[[get:key=value]]`). When retrieving, the guide can limit the key to be within one of the existing values, and will force the value to be the last stored one. Values can be either overridden or added to a set, depending on the domain. This can effectively implement memory, and this can extend beyond a simple context window provided that the guide keeps external memory. In problems like the bAbI tasks Weston et al. (2015) requiring models to keep track of the state of objects and characters through long stories, this guide can reduce the problem of remembering locations (and avoiding interference in self-attention) by the problem of translating each question into a query, using only local information.

**Quote Guide**  Language models often hallucinate quotes, e.g. saying that "According to Wikipedia, 'XYZ'. Therefore...". We can implement a simple `quote` guide that forces quotes to actually come from a trusted source. Specifically, whenever a prefix like `[[quote:]]` is generated, the guide can force the subsequent output to be one of the sentences contained in a certain web page (that set is regular). Externally, an UI can even mark guided quotes with their source, which can be kept by the guide.

**Algebra Guide**  Mathematical reasoning tools can be also integrated as guides for math problem-solving. Peano itself was originally used with algebra problems, and can thus also serve as a guide for mathematical reasoning. We can also leverage other tools, such as computer algebra systems, as guides. One example is the SymPy integration with Codex used previously to solve math word problems He-Yueya et al. (2023), where some instructions can add variables, and some can indicate to the solver a variable to be solved for. In the case of He-Yueya et al. (2023), the model simply indicates which variable to solve for, and the answer is externally extracted. When specifying equations in an `[[eq]]` block, the guide can force the model to output a syntactically valid equation, and also one that only uses already existing variables. This will guarantee that parentheses will be correctly closed (the completion engines in Poesia et al. (2022) achieve this by deriving a completion engine from a parser) and that variables are used after they are introduced. If we wanted the model to also use the results from the solver, we can turn the `[[answer]]` block into a `[[solve:x=v]]` guided block, where $x$ is constrained to be one of the existing variables, and $v$ is given by the algebra solver.

## B    DEONTIQA

We generate the DeontiQA problems following the general methodology of PrOntoQA Saparov & He (2022), where we first sample assumptions and proofs in a logical form and then realize those in natural language. The main qualitative differences are (1) in the specific logical framework we use and (2) in how we translate logical sentences into natural language.

**Logical framework**    We formalize a logical framework for a general domain of managing calendar and event invites in Peano. We create a base type for *actions*, as well as the deontic predicates *permissible* and *obligatory*, to be applied to actions. We have 5 object types: `person`, `entity`, `reminder`, `event` and `invite`. From these, we create 14 kinds of actions:

Context: 1- In a company, there are three employees: Alice, Bob, and Carol. 2- They have a group called "WorkTeam". 3- They have three upcoming events: a team event, a team-building event, and a product launch event. 4- Bob is the organizer of the team-building event. 5- Carol is a participant in the product launch event. 6- The team event is a conference. 7- The team-building event is yearly. 8- The team event is a monthly event and is a short event, while the product launch event is a long event. 9- Alice is busy during the team event, while Bob is free during the product launch event. 10- The team event is a private event, and the product launch event is a public event. 11- If a person is busy during an event, it is impermissible to add them as a participant. 12- If a person is free during an event, it is permissible to add them as a participant. 13- If an event is a long event, it is obligatory to add groups as a participant. 14- If an event has high priority for a person, then it is not obligatory to set a reminder for a few days before the event for that person. 15- If an event is a conference, it is permissible to update the event to be public. 16- If an event has low priority for a person, it is permissible to cancel the event. 17- If an event is short, it is impermissible to reschedule the event yearly. 18- If an event has low priority for a person, then it is obligatory to set a reminder for a few days before the event for that person. 19- If an event is private, it is obligatory to remove Carol as a participant. 20- If a person is a participant in an event, it is permissible to request an event update from them. 21- If a person is the organizer of an event, it is obligatory to change the event's visibility to confidential. 22- If an event is a monthly event, it is a short event. 23- If an event is a yearly event, it is a long event. 24- If an event is long, then Carol has high priority for that event. 25- If an event is a conference, it is a public event. 26- If an event is private, it is a meeting. 27- If an event is public, it is a social event. 28- If a person is the organizer of an event, they are a participant in that event.
Question: Given the rules above, is it not obligatory for Carol to set a reminder for a few days before the team-building event?
Reasoning: The team-building event is a yearly event. The team-building event is a long event. Carol has high priority for the team-building event. If an event has high priority for a person, then it is not obligatory to set a reminder for a few days before the event for that person. Thus, it is not obligatory for Carol to set a reminder for a few days before the yearly team-building event.
Answer (yes or no): Yes, it is not obligatory for Carol to set a reminder for a few days before the yearly team-building event.

Figure 7: Example #1 of a DeontiQA problem.

- Given an event invite, the agent can `accept`, `decline` or `send_notification` for that invite.

- Given an event, the agent can `cancel_event`.

- Given a reminder specification (constructed with a person and a time period before the event), the agent can `set_reminder`.

- Given an event and an entity, the agent can `add_participant` or `remove_participant`.

- Given an event and a person, the agent can `delegate_event`.

- For an event, a person might `request_event_update`, `suggest_alternative_time`, or `check_availability`

- Given an event and a proper event property to be changed, the agent can `update_event`, `reschedule_event`, or `change_visibility`, with the property describing the proper update.

**Problem generation**  To generate a problem, we first sample a theory — a set of hypotheses —, and using those assumptions we try to construct a random derivation (applying axioms and assumptions to hypotheses or previous conclusions). The conclusion of the derivation (or its negation, 50% of the time) becomes the goal in the generated problem.

**Translation to natural language**  To realize the problem in natural language, we use the aid of GPT-4 OpenAI (2023), prompted to translate the logical statements into a story in natural language given a few examples, with sentences describing the axioms. All stories were manually checked to still be unambiguous and logically sound, and we only use GPT-4 to help with diversity. As a result, the DeontiQA problems show greater linguistic variation than both PrOntoQA and ProofWriter, especially at their beginning. We show 3 example problems in Fig. 7, Fig. 8 and Fig. 9. The full set of 60 problems is released along with the supplementary material.

Context: 1- In a marketing agency, there are three employees: Alice, Bob, and Carol. 2- Alice is the marketing manager, Bob is a graphic designer, and Carol is a content writer. 3- The company has a remote team called "CreativeTeam". 4- They have three upcoming events: a brainstorming session, a company party, and a progress event. 5- Alice is the organizer of the brainstorming session, Carol is a participant in the company party, and the CreativeTeam is a group participant in the progress event. 6- The brainstorming session is short, while the company party is a long event. 7- The brainstorming session is a meeting, and the company party is a social event. 8- If an event has high priority for a person, it is permissible for them to suggest an alternative time. 9- If a person is the organizer of an event, it is obligatory to add them as a participant. 10- If a person is free during an event, it is permissible for them to accept an individual invite to the event. 11- If a group is a participant in an event, it is permissible to delegate the event to the group. 12- If a person is busy during an event, it is impermissible to set a reminder for a few minutes before the event. 13- If a person is a participant in an event, it is permissible to remove them as a participant. 14- For short events, it is permissible to update the event to a social event. 15- If a person's status is tentative for an event, it is obligatory to check the availability of the person for that event. 16- If an event has high priority for a person, it is obligatory to set a reminder for a few days before the event. 17- If a person is a participant in an event, it is impermissible for them to suggest an alternative time. 18- If an event is public, it is obligatory to add Alice as a participant. 19- Meetings are public events. 20- Public events are short events. 21- If a person is the organizer of an event, their priority for that event is high. 22- If a person is a participant in an event, they are available for that event. 23- If an event is short, Bob is a participant. 24- Daily events are short events. 25- If a person has a high priority for an event, they are busy during that event. 26- If a person has a low priority for an event, they are free during that event. 27- If a group is a participant in an event, the event is public. Question: Given the rules above, is it permissible for Bob to suggest an alternative time for the progress event? Reasoning: The CreativeTeam is a group participant in the progress event. The progress event is a public event. The progress event is a short event. Bob is a participant in the progress event. It is impermissible for a participant to suggest an alternative time for an event they are participating in. Answer (Yes or no): No

Figure 8: Example #2 of a DeontiQA problem.

Context: 1- In a software company, there are three project managers: Alice, Bob, and Carol. 2- They have a team called "DevTeam". 3- They have two upcoming events: a team-building activity and a project review event. 4- Alice is the organizer of the team-building activity, Bob is a participant in the team-building activity, and the entire DevTeam is participating in the team-building activity. 5- The project review is a short event. 6- The team-building activity is a social event. 7- The team-building activity is a public event. 8- If a person is the organizer of an event, it is obligatory to add them as a participant. 9- If a person is a participant in an event, it is permissible for them to accept an individual invite to the event. 10- If an event is short, it is impermissible to cancel the event. 11- If a group is participating in an event, it is obligatory to add the group as a participant. 12- If a person is free during an event, it is permissible to set a reminder for a few hours before the event. 13- If a person is busy during an event, it is impermissible to reschedule the event to be a daily event. 14- If an event is public, it is obligatory to check Alice's availability for the event. 15- If a person is a participant in an event, it is permissible to delegate the event organization to them. 16- If a person's availability for an event is tentative, it is permissible for them to suggest an alternative time for the event. 17- If an event has high priority for a person, then it is obligatory to set a reminder for them for a few days before the event. 18- If a person's availability for an event is free, it is impermissible for them to suggest an alternative time for the event. 19- If an event is short, then it is a meeting. 20- If an event is a meeting, then Bob's availability for the event is tentative. 21- If Alice's availability for an event is tentative, then she is a participant in the event. 22- If a person is free for an event, then the event has low priority for them. 23- If an event is public, then it is a social event. 24- If an event is private, then it is a personal event. 25- If an event is daily, then it is not a weekly event. 26- If an event is weekly, then it is not a monthly event. Question: Given the rules above, is it permissible for Bob to suggest an alternative time for the project review? Reasoning: The project review is a meeting. Bob's availability for the project review is tentative. It is permissible for a person with tentative availability to suggest an alternative time for the event. Therefore, it is permissible for Bob to suggest an alternative time for the project review. Answer (Yes or no): Yes, it is permissible for Bob to suggest an alternative time for the project review.

Figure 9: Example #3 of a DeontiQA problem.

## C Constrained Semantic Decoding with Chat Models

Originally, Constrained Semantic Decoding was proposed to work with standard autoregressive language models Poesia et al. (2022). This relied on the ability to bias the logits of the model during generation, which is both possible in local models as well as through the OpenAI API[2]. The OpenAI gpt3.5-turbo has a different API, since it is a chat-based model. In this API, we pass a list of messages with roles (user or assistant, where the model understands the latter as marking its own past generations). The API also has a logit bias parameter. However, we unfortunately cannot pass an incomplete prefix for the model's response. Thus, we are unable to force the model to complete a certain message while also applying logit biases. Every completion starts a new message. This requires an adaptation to the procedure in Poesia et al. (2022).

We start with the usual rejection-based CSD procedure: we put few-shot examples in the previous messages showcasing the response format we want, and sample a full response from the model. We then use token-by-token CSD to validate the response. If this terminates without finding any violation, we're done — the entire generation, including choices made in guided blocks (e.g., infer), were valid. If not, like in original CSD, we take the largest valid prefix and use the CSD algorithm to compute the set of tokens that are valid after that prefix.

Here we reach the main difference in the API. We want the model to continue its message from the last valid token. However, this is not possible in the current API. Instead, we must request a new message. Fortunately, we found gpt3.5-turbo to very frequently simply continue its generation when its last message appears incomplete[3]. We exploit this behavior and (1) request a new message with a single token, passing the set of valid tokens in the logit bias, (2) append the sampled token to the previous message and request a new, unconstrained message, and (3) repeat until we have received a full response.

When the model continues from where it stops, this approach is essentially equivalent to rejection-based CSD. Unfortunately, it is not fully reliable. In a number of cases, we found the model to insistently apologize after noticing that its previous message was incomplete. This is problematic when the previous message started a guided block that is to be finished in the next message. In this case, the model's apology is contained in the guided block, and is thus always an invalid generation. What happens is that this immediately triggers a violation, and the CSD procedure above is executed.

Often, the CSD corrections will eventually get the model to make enough choices to complete the guided block, at which point its apology is not an issue anymore (e.g., see Fig. 10. In rare ($< 0.1\%$) cases, the issue persists and we cannot recover from the apology (Fig. 11 shows an example). To avoid a prohibitive number of API calls, we aborted sampling when more than 20 violations were hit in the same solution.

## D Experimental Details

Experiments with the OpenAI models were made using their public API. For LLaMA 13B, we ran and fine-tuned the model on an NVIDIA A100 80GB GPU. For fine-tuning when running STaR (§4.3), we performed inference on 200 problems — 40 for each number of hops from 1 to 5 — in each STaR iteration, and collected the generations where the model reached the correct answer (with each of the 3 criteria described in §4.3). We fine-tuned for 1 epoch (i.e., seeing each example exactly once) with a batch size of 2 and a learning rate of 2e-5. We used the Adam8bit optimizer with default parameters, reset in each iteration of STaR.

## E Prompts

All of our prompts are provided in the attached supplementary material. We use JSON files for the chat model prompts, exactly as we pass them to the OpenAI API.

---

[2]https://platform.openai.com/docs/api-reference/completions/create#completions/create-logit_bias

[3]We hypothesize this is done so that models also complete their messages when the token limit is hit in the OpenAI Playground and users immediately request a new completion

Formalized context: 1- Every [[prop:dumpus]] is an [[prop:impus]]. [[axiom:(dumpus 'x) -> (impus 'x)]]. 2- Each [[prop:vumpus]] is not [[prop:luminous]]. [[axiom:(vumpus 'x) -> (not (luminous 'x))]]. 3- [[prop:dumpus]] are [[prop:orange]]. [[axiom:(dumpus 'x) -> (orange 'x)]]. 4- Every [[prop:wumpus]] is [[prop:bitter]]. [[axiom:(wumpus 'x) -> (bitter 'x)]]. 5- Each [[prop:jompus]] is not [[prop:orange]]. [[axiom:(jompus 'x) -> (not (orange 'x))]]. 6- [[prop:wumpus]] are [[prop:numpus]]. [[axiom:(wumpus 'x) -> (numpus 'x)]]. 7- [[prop:impus]] are [[prop:rompus]]. [[axiom:(impus 'x) -> (rompus 'x)]]. 8- Every [[prop:impus]] is [[prop:opaque]]. [[axiom:(impus 'x) -> (opaque 'x)]]. 9- [[prop:numpus]] are [[prop:dull]]. [[axiom:(numpus 'x) -> (dull 'x)]]. 10- [[prop:vumpus]] are [[prop:wumpus]]. [[axiom:(vumpus 'x) -> (wumpus 'x)]]. 11- Each [[prop:numpus]] is a [[prop:dumpus]]. [[axiom:(numpus 'x) -> (dumpus 'x)]]. 12- [[object:wren]] is a [[prop:dumpus]]. [[axiom:(dumpus wren)]]. Formalized goal: [[goal:(orange wren)]]
Reasoning: [[infer:(impus wren)]] Wren is an impus. [[infer:(rompus wren)]]'Apologies for the incomplete message.
Continuing from where I stopped: [[infer:(opaque wren)]]"My apologies for the incomplete message.
Continuing from where I stopped: [[infer:(orange wren)]]] Wren is orange. This was the goal.
Answer: True

Figure 10: Example generation of `gpt3.5-turbo` with LOGICGUIDE on PrOntoQA where the model started to apologize twice after violating the guide, but it eventually recovered with our chat adaptation of CSD.

Formalized context: 1- [[object:bob]] is [[prop:blue]]. [[axiom:(blue bob)]]. 2- [[object:bob]] is [[prop:quiet]]. [[axiom:(quiet bob)]]. 3- [[object:bob]] is [[prop:red]]. [[axiom:(red bob)]]. 4- [[object:bob]] is [[prop:rough]]. [[axiom:(rough bob)]]. 5- [[object:bob]] is [[prop:smart]]. [[axiom:(smart bob)]]. 6- [[object:charlie]] is [[prop:cold]]. [[axiom:(cold charlie)]]. 7- [[object:charlie]] is [[prop:red]]. [[axiom:(red charlie)]]. 8- [[object:dave]] is [[prop:blue]]. [[axiom:(blue dave)]]. 9- [[object:dave]] is [[prop:quiet]]. [[axiom:(quiet dave)]]. 10- [[object:erin]] is [[prop:cold]]. [[axiom:(cold erin)]]. 11- [[object:erin]] is [[prop:white]]. [[axiom:(white erin)]]. 12- If something is [[prop:smart]] then it is [[prop:white]]. [[axiom:(smart 'x) -> (white 'x)]]. 13- All [[prop:quiet]], [[prop:rough]] things are [[prop:smart]]. [[axiom:(ifsomething isPSSorryPSI apologizePSI apologizePSI apologizePSI apologizePSI apologizePSI apologizePSI apologizePSI apologizePSI apologizePSI apologizePSI apologizePSI apologizePSFormalized contextPSI apologize for the previous messagePSIPSIPSI apologize for the previous messagePSI

Figure 11: Example generation of `gpt3.5-turbo` with LOGICGUIDE on ProofWriter where the model started to apologize after violating the guide, and did not recover. Here, it violated the s-expr syntax for defining axioms in Peano, but after being constrained it still insists in apologizing, instead of trying to finish the formula. This behavior is rare ($< 0.1\%$ of the cases), and would be avoided if the API had a parameter to give the model an incomplete message for it to complete.

Context: Every dog is small. Every feline is a snake. Every animal is not bitter. Sheep are bitter. Cats are carnivores. Each vertebrate is a mammal. Mammals are felines. Each vertebrate is dull. Snakes are cats. Cats are not kind. Every snake is not happy. Every sheep is a vertebrate. Each feline is cold. Each dog is a sheep. Every mammal is not liquid. Every carnivore is a cow. Every carnivore is brown. Alex is a sheep. Question: True or false: Alex is not bitter.

Formalized context: 1- Every [[prop:dog]] is [[prop:small]]. [[axiom:(dog 'x) -> (small 'x)]]. 2- Every [[prop:feline]] is a [[prop:snake]]. [[axiom:(feline 'x) -> (snake 'x)]]. 3- Every [[prop:animal]] is not [[prop:bitter]]. [[axiom:(animal 'x) -> (not (bitter 'x))]]. 4- [[prop:sheep]] are [[prop:bitter]]. [[axiom:(sheep 'x) -> (bitter 'x)]]. 5- [[prop:cat]] are [[prop:carnivore]]. [[axiom:(cat 'x) -> (carnivore 'x)]]. 6- Each [[prop:vertebrate]] is a [[prop:mammal]]. [[axiom:(vertebrate 'x) -> (mammal 'x)]]. 7- [[prop:mammal]] are [[prop:feline]]. [[axiom:(mammal 'x) -> (feline 'x)]]. 8- Each [[prop:vertebrate]] is [[prop:dull]]. [[axiom:(vertebrate 'x) -> (dull 'x)]]. 9- [[prop:snake]] are [[prop:cat]]. [[axiom:(snake 'x) -> (cat 'x)]]. 10- [[prop:cat]] are not [[prop:kind]]. [[axiom:(cat 'x) -> (not (kind 'x))]]. 11- Every [[prop:snake]] is not [[prop:happy]]. [[axiom:(snake 'x) -> (not (happy 'x))]]. 12- Every [[prop:sheep]] is a [[prop:vertebrate]]. [[axiom:(sheep 'x) -> (vertebrate 'x)]]. 13- Each [[prop:feline]] is [[prop:cold]]. [[axiom:(feline 'x) -> (cold 'x)]]. 14- Each [[prop:dog]] is a [[prop:sheep]]. [[axiom:(dog 'x) -> (sheep 'x)]]. 15- Every [[prop:mammal]] is not [[prop:liquid]]. [[axiom:(mammal 'x) -> (not (liquid 'x))]]. 16- Every [[prop:carnivore]] is a [[prop:cow]]. [[axiom:(carnivore 'x) -> (cow 'x)]]. 17- Every [[prop:carnivore]] is [[prop:brown]]. [[axiom:(carnivore 'x) -> (brown 'x)]]. 18- [[object:alex]] is a [[prop:sheep]]. [[axiom:(sheep alex)]].

Formalized goal: [[goal:(not (bitter alex))]]

Reasoning: [[infer:(vertebrate alex)]] Alex is a vertebrate. [[infer:(dull alex)]] Alex is dull. [[infer:(mammal alex)]] Alex is a mammal. [[infer:(feline alex)]] Alex is a feline. [[infer:(not (liquid alex))]] Alex is not liquid. [[infer:(snake alex)]] Alex is a snake. [[infer:(cat alex)]] Alex is a cat. [[infer:(carnivore alex)]] Alex is a carnivore. [[infer:(not (kind alex))]] Alex is not kind. [[infer:(not (happy alex))]] Alex is not happy. [[infer:(cold alex)]] Alex is cold. [[infer:(brown alex)]] Alex is brown. [[infer:(cow alex)]] Alex is a cow. [[infer:(bitter alex)]] Alex is bitter. This contradicts the goal.

Answer: False.

Figure 12: Example generation of `text-davinci-003` with LOGICGUIDE on the PrOntoQA False Ontology. The solution to this problem can be reasoned in one step, but the model is strongly biased towards making wrong conclusions. LOGICGUIDE avoids those, but the model still takes many more inference steps to eventually reach the goal, deriving many useless facts first.

# F COMPLETE SAMPLES - PRONTOQA/PROOFWRITER

We here provide full samples of solutions generated by language models with LOGICGUIDE, also showcasing the most common failure modes.

Fig. 12 shows one case of `text-davinci-003` on the PrOntoQA False Ontology, where the model properly formalizes all of the assumptions, but still tries to make wrong conclusions very often. As a result, its solution ends up taking a long detour to eventually get to the goal, but eventually does so correctly (it can be concluded directly from two of the assumptions).

Fig. 14 shows one example of `gpt3.5-turbo` on ProofWriter, where the model further justifies its solution based on the axioms. We found these post-hoc justifications to be highly accurate. Unguided models sometimes also justify their inferences even if not prompted to do so, but to do so they must procuce hallucinations (or assume general world knowledge, such that "an animal cannot chase itself").

Fig. 13 shows one rare failure mode where the model misclassifies whether it has already proved the goal, and thus does not proceed further. We can detect this failure mode with LOGICGUIDE, since we have access to the Peano state and can ask the environment whether the goal was proved or not. In this way, as explained in App. D, we can distinguish certified and uncertified answers.

Fig. 15 shows a case where LLaMA 13B misformalized (several) assumptions, whereas Fig. 16 shows a similar case with `text-davinci-003` (much more rare). The result in both cases is that the model cannot make progress in its formal inferences, instead of making invalid deductions. Again, since we can detect when the answer was not formally derived, we can avoid fine-tuning on these cases where the model still guesses the right answer but with unsond reasoning, as we exploited in §4.3.

Formalized context: 1- The [[object:cat]] is [[prop:red]]. [[axiom:(red cat)]]. 2- The [[object:cat]] [[relation:visits]] the [[object:cow]]. [[axiom:(visits cat cow)]]. 3- The [[object:cow]] is [[prop:big]]. [[axiom:(big cow)]]. 4- The [[object:cow]] [[relation:needs]] the [[object:dog]]. [[axiom:(needs cow dog)]]. 5- The [[object:cow]] [[relation:needs]] the [[object:squirrel]]. [[axiom:(needs cow squirrel)]]. 6- The [[object:dog]] does not [[relation:need]] the [[object:cat]]. [[axiom:(not (needs dog cat))]]. 7- The [[object:dog]] [[relation:visits]] the [[object:cow]]. [[axiom:(visits dog cow)]]. 8- The [[object:squirrel]] [[relation:chases]] the [[object:cow]]. [[axiom:(chases squirrel cow)]]. 9- The [[object:squirrel]] is [[prop:nice]]. [[axiom:(nice squirrel)]]. 10- The [[object:squirrel]] [[relation:needs]] the [[object:dog]]. [[axiom:(needs squirrel dog)]]. 11- If someone [[relation:needs]] the [[object:squirrel]] then they [[relation:chase]] the [[object:cat]]. [[axiom:(needs 'x squirrel) -> (chases 'x cat)]]. 12- If someone [[relation:chases]] the [[object:cat]] then the [[object:cat]] [[relation:visits]] the [[object:cow]]. [[axiom:(chases 'x cat) -> (visits cat cow)]]. 13- If someone [[relation:chases]] the [[object:cat]] then they are [[prop:nice]]. [[axiom:(chases 'x cat) -> (nice 'x)]]. 14- If someone [[relation:chases]] the [[object:squirrel]] then the [[object:squirrel]] [[relation:chases]] the [[object:cow]]. [[axiom:(chases 'x squirrel) -> (chases squirrel cow)]]. 15- If someone [[relation:chases]] the [[object:cow]] and they [[relation:needs]] the [[object:squirrel]] then the [[object:cow]] [[relation:needs]] the [[object:squirrel]]. [[axiom:(chases 'x cow) -> (needs cow squirrel) -> (needs cow squirrel)]]. 16- If someone is [[prop:nice]] then they do not [[relation:need]] the [[object:cat]]. [[axiom:(nice 'x) -> (not (needs 'x cat))]]. 17- If someone [[relation:needs]] the [[object:dog]] and they are [[prop:red]] then they [[relation:chase]] the [[object:dog]]. [[axiom:(needs 'x dog) -> (red 'x) -> (chases 'x dog)]]. 18- If someone is [[prop:nice]] and not [[prop:green]] then they do not [[relation:visit]] the [[object:cat]]. [[axiom:(nice 'x) -> (not (green 'x)) -> (not (visits 'x cat))]].
Formalized goal: [[goal:(not (needs cow cat))]]
Reasoning: [[infer:(chases cow cat)]] The cow does not chase the cat. [[infer:(not (needs squirrel cat))]]] The cow does not need the cat. This was the goal.
Answer: True

Figure 13: Example generation of `gpt3.5-turbo` with LOGICGUIDE on ProofWriter where the model misidentifies whether it has reached the goal or not. Given the Peano state and the formalized goal, we can at any point detect whether the model has indeed proved its goal or not, i.e. whether its answer was certified or not. This can likely be avoided by making the step of outputting a conclusion be prefixed with a formal step, which can be constrained to only be allowed once the goal has been proved or disproved.

(...) Formalized goal: [[goal:(quiet gary)]]
Reasoning: [[infer:(young gary)]] Gary is young because he is round (from axiom 9). [[infer:(white gary)]] Gary is white because he is round and all round people are white (from axiom 5). [[infer:(not (quiet gary))]]] Gary is not quiet because he is young and white, and all young, white people are not quiet (from axiom 7). Therefore, the answer is False.

Figure 14: Example generation of `gpt3.5-turbo` with LOGICGUIDE on ProofWriter, where the model provides further justification on the inferences. We found that models are highly accurate in justifying the inferences post-hoc, provided that they are correct.

Formalized context: 1- The [[object:lion]] chases the [[object:squirrel]]. [[axiom:(chases lion squirrel)]]. 2- The [[object:lion]] is [[prop:big]]. [[axiom:(big lion)]]. 3- The [[object:lion]] is [[prop:blue]]. [[axiom:(blue lion)]]. 4- The [[object:lion]] is [[prop:kind]]. [[axiom:(kind lion)]]. 5- The [[object:lion]] is [[prop:nice]]. [[axiom:(nice lion)]]. 6- The [[object:lion]] is [[prop:young]]. [[axiom:(young lion)]]. 7- The [[object:lion]] needs the [[object:squirrel]]. [[axiom:(needs lion squirrel)]]. 8- The [[object:lion]] visits the [[object:squirrel]]. [[axiom:(visits lion squirrel)]]. 9- The [[object:squirrel]] chases the [[object:lion]]. [[axiom:(chases squirrel lion)]]. 10- The [[object:squirrel]] is [[prop:big]]. [[axiom:(big squirrel)]]. 11- The [[object:squirrel]] is [[prop:blue]]. [[axiom:(blue squirrel)]]. 12- The [[object:squirrel]] is [[prop:kind]]. [[axiom:(kind squirrel)]]. 13- The [[object:squirrel]] is [[prop:nice]]. [[axiom:(nice squirrel)]]. 14- The [[object:squirrel]] is [[prop:young]]. [[axiom:(young squirrel)]]. 15- The [[object:squirrel]] needs the [[object:lion]]. [[axiom:(needs squirrel lion)]]. 16- The [[object:squirrel]] visits the [[object:lion]]. [[axiom:(visits squirrel lion)]]. 17- If someone visits the [[object:squirrel]] then they chase the [[object:lion]]. [[axiom:(visits squirrel lion) -> (chases lion)]]
Formalized goal: [[goal:(chases lion lion)]]
Reasoning: [[infer:nothing]] The lion does not chase the lion. This contradicts the goal.
Answer: False

Figure 15: Example generation of LLaMA 13B with LOGICGUIDE on ProofWriter, where the model misformalizes assumption 17. The natural language sentence reads "if someone visits the squirrel, then they chase the lion", but the logical formulation written by the model would translate to "if the squirrel visits the lion, then the lion chases". As a result, the model is not able to make any formal inferences. It still outputs an uncertified answer, which we can detect.

(...) 13- If something [[relation:visits]] the [[object:squirrel]] and the [[object:squirrel]] [[relation:needs]] the [[object:bald_eagle]] then the [[object:squirrel]] is [[prop:green]]. [[axiom:(visits 'x squirrel) -> (needs squirrel bald_eagle) -> (green 'x)]]. 14- If something [[relation:needs]] the [[object:squirrel]] then it is [[prop:green]]. [[axiom:(needs 'x squirrel) -> (green 'x)]].
Formalized goal: [[goal:(green squirrel)]]
Reasoning: [[infer:(green bald_eagle)]] The bald eagle is green. [[infer:(green cow)]] The cow is green. [[infer:nothing]] Nothing can be inferred from the context about the squirrel. This contradicts the goal.
Answer: False.

Figure 16: Example generation of `text-davinci-003` with LOGICGUIDE on ProofWriter, where the model misformalizes assumption 13. The natural language sentence reads "if someone visits the squirrel and the squirrel needs the bald eagle, then the squirrel is green", but the logical formulation written by the model would translate to "if someone visits the squirrel and the squirrel needs the bald eagle, then *they* are green" (not the squirrel is green). Like in Fig. 15, the model is not able to make any formal inferences after its second step, and we can detect that its answer was not certified.

Patient: Pharmacists maintain that doctors should not be permitted to sell the medicine that they prescribe because doctors would then be tempted to prescribe unnecessary medicines in order to earn extra income. But pharmacists have a financial interest in having a monopoly on the sale of prescription medicines, so their objection to the sale of medicines by doctors cannot be taken seriously. The patient's argument proceeds by
A. attempting to discredit a position by questioning the motives of the proponents of that position
B. rejecting a questionable position on the grounds that the general public does not support that position
C. pointing out an unstated assumption on which the pharmacists' argument relies and then refuting it
D. asserting that pharmacists lack the appropriate knowledge to have informed opinions on the subject under discussion
Answer: A

**gpt-3.5-turbo:** In this problem, the patient presents an argument against the objection raised by pharmacists regarding doctors selling medicines. The patient claims that the objection cannot be taken seriously because pharmacists have a financial interest in maintaining a monopoly on the sale of prescription medicines.
To determine how the patient's argument proceeds, let's examine each option:
A. The patient's argument does question the motives of the pharmacists by suggesting that they have a financial interest in the matter. However, it does not discredit the entire position of the pharmacists based solely on their motives. Therefore, option A is not the correct choice.
B. The patient's argument does not rely on the fact that the general public does not support the pharmacists' position. Instead, it focuses on the financial interest of the pharmacists. Therefore, option B is not the correct choice.
C. The patient's argument does not explicitly point out an unstated assumption of the pharmacists' argument and then refute it. Instead, it directly challenges the motives of the pharmacists by highlighting their financial interest. Therefore, option C is not the correct choice.
D. The patient's argument does not discuss the knowledge of pharmacists or their opinions on the subject under discussion. Therefore, option D is not the correct choice.
After considering all the options, we can conclude that none of the given options accurately describes how the patient's argument proceeds.
Answer: None of the options.
**gpt-3.5-turbo + Base Bootstrap**: Reasoning: The patient argues that the objection raised by pharmacists is not valid because they have a financial interest in maintaining a monopoly on the sale of prescription medicines. Answer: A. attempting to discredit a position by questioning the motives of the proponents of that position
**gpt-3.5-turbo + LogicGuide Bootstrap:** Reasoning: The patient argues that the objection of pharmacists to doctors selling medicine cannot be taken seriously because pharmacists have a financial interest in maintaining a monopoly on the sale of prescription medicines. Answer: A. attempting to discredit a position by questioning the motives of the proponents of that position

Figure 17: Example #1 of a ReClor problem. Here, the base model does a very long analysis that ultimately discards option A, by questioning whether it "discredited the entire position of the pharmacist", which is irrelevant to the question. Bootstrapped models recognize the form of the argument as sufficiently matching option A.

# G    COMPLETE SAMPLES - RECLOR

We here show three full examples of questions and answers from the ReClor dataset, highlighting the differences in the base GPT-3.5 Turbo model and both of its bootstrapped versions. The zero-shot prompt we used was: *"Answer the problem below. First, carefully analyze the context and each of the options before deciding which is correct. Then, at the end of your response, end with a line with the format "Answer: X", indicating one of the options."*. We tested several variations of these instructions, picking the one where the base model obeyed the expected format the best.

Traveler: Southern Airways has a far worse safety record than Air Dacentaria over the past few years, in terms of both the frequency of accidents and the severity of accidents. Ttherefore, although Air Dacentaria is slightly more expensive, I will choose it over Southern Airways for my flight from Pederton to Dacenta, since it is worth paying extra to have a safer flight.

Which of the following, if true, casts the most serious doubt on the traveler's argument?

A. Although the frequency of accidents is greater on Southern Airways, on both airlines the proportion of flights that have accidents is very small.

B. Most Southern Airways flights, but not its flight from Pederton to Dacenta, use Sarmouth airport, which because of its mountainous location is one of the world's most dangerous.

C. Only in the last few years has the cost of flying from Pederton to Dacenta been more expensive on Air Dacentaria than on Southern Airways.

D. Air Dacentaria's flight from Pederton to Dacenta is nonstop, whereas Southern Airways' flight touches down in Gorinda on the way.

Answer: B

**gpt-3.5-turbo** Let's analyze each of the options:

Option A states that although the frequency of accidents is greater on Southern Airways, the proportion of flights that have accidents is very small on both airlines. If this is true, it diminishes the significance of the difference in safety records between the two airlines, as the overall likelihood of being in an accident is small for both.

Option B claims that most Southern Airways flights, except for the one from Pederton to Dacenta, use Sarmouth airport, which is considered one of the world's most dangerous due to its mountainous location. This information may suggest that the higher accident frequency of Southern Airways might be due to its use of a particularly hazardous airport. However, since the traveler is specifically referring to the flight from Pederton to Dacenta, this option may not significantly impact their argument.

Option C states that only in the last few years has Air Dacentaria become more expensive than Southern Airways for flying from Pederton to Dacenta. This suggests that the increased cost of Air Dacentaria might be a recent change, and the traveler's decision may have been based on outdated information or prior pricing structures.

Option D mentions that Air Dacentaria's flight from Pederton to Dacenta is nonstop, while Southern Airways' flight makes a stop in Gorinda. If this is true, it may imply that the nonstop flight on Air Dacentaria would reduce the chances of potential accidents compared to the Southern Airways flight that involves a touch-down in Gorinda.

After analyzing the options, it appears that option D, which states that Air Dacentaria's flight from Pederton to Dacenta is nonstop, while Southern Airways' flight touches down in Gorinda, casts the most serious doubt on the traveler's argument. This information suggests that the nonstop flight on Air Dacentaria could potentially offer a safer flight experience even with the additional cost.

Answer: D

**gpt-3.5-turbo + Base Bootstrap** Reasoning: The traveler prioritizes safety over cost. Air Dacentaria is slightly more expensive but safer. A. This does not cast doubt on the traveler's argument, as it still supports the idea that Air Dacentaria is safer. B. This is irrelevant to the safety of Air Dacentaria and Southern Airways. C. This does not cast doubt on the traveler's argument, as it still supports the idea of choosing Air Dacentaria for safety reasons. D. This supports the traveler's argument, as it highlights that Air Dacentaria's flight is more convenient and potentially safer. Answer: D

**gpt-3.5-turbo + LogicGuide Bootstrap** Reasoning: The traveler's argument is based on the safety record of the airlines. To cast doubt on the argument, we need to find information that undermines the relevance of the safety record. A. This information does not undermine the traveler's argument, as the proportion of accidents is small on both airlines. B. This information weakens the traveler's argument, as the specific flight from Pederton to Dacenta is not affected by the dangerous airport. C. This information does not undermine the traveler's argument, as the cost comparison is only relevant to the decision-making process. D. This information does not undermine the traveler's argument, as nonstop flights are generally considered safer. Answer: B

Figure 18: Example #2 of a ReClor problem. Here, both the base model and the base bootstrapped model ignore the key connection between option B and the traveler's argument, which the LogicGuide-boostrapped model picks up on..

Dario: The government should continue to grant patents for all new drug compounds. Patents promote innovation by rewarding pharmaceutical companies for undertaking the costly research needed to develop new drugs. Cynthia: Promoting innovation is certainly important. For this reason, patents should be granted only for truly innovative drugs, not for minor variants of previously existing drugs. Since it is much less expensive to tweak an existing drug than to develop a wholly new one, pharmaceutical companies tend to focus primarily on the cheaper alternative.

Dario and Cynthia disagree over whether

A. patents should be granted for all drug compounds

B. developing truly innovative drugs is costly

C. drug patents can promote innovation

D. pharmaceutical companies have an incentive to create minor variants of existing drugs

Answer: A

**gpt-3.5-turbo:** In this argument, Dario believes that the government should continue to grant patents for all new drug compounds because patents promote innovation by rewarding pharmaceutical companies for the costly research needed to develop new drugs. On the other hand, Cynthia argues that patents should only be granted for truly innovative drugs, not for minor variants of previously existing drugs. She points out that it is much less expensive to tweak an existing drug than to develop a wholly new one, and pharmaceutical companies tend to focus primarily on the cheaper alternative.

Analyzing the options: A. The disagreement between Dario and Cynthia is not about whether patents should be granted for all drug compounds. Dario supports this idea while Cynthia argues for granting patents only for truly innovative drugs. Therefore, option A can be eliminated. B. The disagreement between Dario and Cynthia is not about whether developing truly innovative drugs is costly. They both agree that it requires costly research. Therefore, option B can be eliminated. C. The disagreement between Dario and Cynthia is about whether drug patents can promote innovation. Dario believes that patents promote innovation, but Cynthia argues that they should only be granted for truly innovative drugs to effectively promote innovation. This aligns with the subject of their disagreement. Therefore, option C is a valid choice. D. The disagreement between Dario and Cynthia is not about whether pharmaceutical companies have an incentive to create minor variants of existing drugs. Both agree that it is a cheaper alternative, and Cynthia specifically mentions this as a concern. Therefore, option D can be eliminated.

Answer: C. drug patents can promote innovation

**gpt-3.5-turbo + Base Bootstrap** Reasoning: Cynthia agrees that patents promote innovation but thinks they should be granted only for truly innovative drugs, not for minor variants of existing drugs. Answer: D. pharmaceutical companies have an incentive to create minor variants of existing drugs

**gpt-3.5-turbo + LogicGuide Bootstrap** Reasoning: Dario supports granting patents for all new drug compounds, while Cynthia believes patents should be granted only for truly innovative drugs, not for minor variants of previously existing drugs. Answer: C. drug patents can promote innovation

Figure 19: Example #3 of a ReClor problem. Here, all models choose incorrect options. The model with LogicGuide bootstrap does seem to pick up the key point of disagreement, but still fails to pick the option that best describes it.

