# OpenReview forum: "Certified Deductive Reasoning with Language Models"
_ICLR.cc/2024/Conference — Submitted to ICLR 2024_

### Official Review · Reviewer_GDuS · 2023-10-18

**Soundness:** 3 good
**Presentation:** 3 good
**Contribution:** 4 excellent
**Rating:** 8
**Confidence:** 3

**Summary:**

This paper proposed a way to utilize a theorem prover with a large language model to produce answers.

**Strengths:**

The high level idea seems good (but the details I'm no so clear about). The results are very good.

**Weaknesses:**

The main problem with this that the details of the architecture isn't clear. Here is what I understand: The LLM gets language (The "Context" in the figures). The LLM generates a "formalized context" that can be used as the input to Peano. Peano implements a guide function, and outputs a set of valid one-step conclusions. This is input back into the LLM by biasing the logits (whatever that means), then presumably the LLM does sometime else to generate the next formalized contexts to do the next steps and so on. At some stage this halts and one of them produces an answer. (Does the LLM also outputs natural language?)
[Alternatively: Using figure 2 as an example, The LLM takes the contact and produces the formalized context and the formalized goal. Peano takes these and outputs a proof (Is this the "reasoning" in that figure?). That would seem to make the most sense. But that can't be correct as the external tool only answers "what inferences can be made next?".]

**Questions:**

What is the interface between the LLM and Peano? (What is the input of each and what is the output? Does Peano have any knowledge built-in (e.g., axioms for deontic logic)?

What is an example application beyond artificial logic puzzles? (The legal reasoning is a good example, but it only used the theorem prover for bootstrapping.)

What does "bias the logits" mean? How is it done? How does the theorem prover determine how to bias them?

(My rating assumes there is a satisfactory answer to these questions. I will downgrade my rating if I still cannot understand the interface after the rebuttal period.)

---

> ### Author Response · Authors · 2023-11-15
> **Author response**
>
> Thank you for the review and the encouraging comments! We provide clarifications on the approach and other questions below, and are happy to discuss any of these further.
>
> > What is the interface between the LLM and Peano? (What is the input of each and what is the output?
>
> We're happy to expand on the explanation in Section 3.1 (we'll update the Appendix with pseudocode based on the description below and the discussion here). Essentially, three components that are tightly integrated: the LLM decoding algorithm, LogicGuide, and Peano. The LLM decoding algorithm will output token by token while constraining the output for the next token to come from a subset of the vocabulary determined by LogicGuide. Outside of a guided block (delimited by "[[" and "]]"), there are no constraints on which token can be emitted. When the model opens a guided block with "[[", LogicGuide first constrains it to output one of the LogicGuide actions followed by a colon. In the cases where the model outputs the "infer" action, LogicGuide calls Peano to determine which inferences are valid at this point. Peano will take as input all the formalized premises and previous inferences, which are extracted from the LLM's output so far, and output a list of valid one-step inferences. LogicGuide will take this list and compute token constraints for the LLM decoding algorithm so that its output, even over multiple tokens, will come from this list. Thus, looking at Figure 2, all three components are tightly connected to produce the output. The input here for the model is just the Context and Question. The model then starts to write its "Formalized context", then the goal, and finally the reasoning and answer. Every time the model opens a block with "[[", LogicGuide is called to determine constraints. In particular, in the [[infer]] blocks, LogicGuide calls Peano to determine valid inferences.
>
> In general, applying such constraints is non-trivial because LLM tokens can break up these sequences (such as "[[", where the brackets might be split up in separate BPE tokens depending on the surrounding context). These complications are handled by CSD, which we use for decoding with constraints.
>
> > Does Peano have any knowledge built-in (e.g., axioms for deontic logic)?
>
> In our experiments, we initialized Peano with an empty theory in PrOntoQA, ProofWriter and DeontiQA -- the relevant axioms are formalized by the LLM in-context. However, it might also be interesting for future work to explore scenarios where some background knowledge is given to the language model implicitly through the theory used to initialize Peano.
>
> > What is an example application beyond artificial logic puzzles? (The legal reasoning is a good example, but it only used the theorem prover for bootstrapping.)
>
> The framework of guides can generally serve to explore connecting formal systems with LLM reasoning, besides the examples we provided. Many other important fragments of reasoning have been formalized into different logics - temporal logic (e.g., for reasoning about events), modal logic (e.g., connected to theory of mind reasoning), deontic logic (e.g., permissions, obligations), and so on. We provide a starting point for these, since these logics can be formalized in Peano and thus used to guide LLMs.
>
> > What does "bias the logits" mean? How is it done? How does the theorem prover determine how to bias them?
>
> The output layer of the Transformer will predict unnormalized weights (logits) for each token in its vocabulary at each iteration. These logits are passed through a softmax to then become a probability distribution for sampling. When doing constrained decoding, such as in CSD, we want to "select" a subset of the tokens that will be allowed for sampling the next token. In some APIs like the OpenAI API, there is no option for directly selecting vocabulary tokens, but there is a way to add a "logit bias" - a weight that will be added before softmax to the output logit of the given vocabulary tokens. A sufficiently large negative bias effectively sets the probability for that token to zero. This trick, which was used in the CSD paper [1], is how we effectively constrain the OpenAI LLMs to valid completions. We use the same trick with LLaMA offline.
>
> > (My rating assumes there is a satisfactory answer to these questions. I will downgrade my rating if I still cannot understand the interface after the rebuttal period.)
>
> That is fair – we will be happy to answer further questions or try to explain in different ways! Let us know what would be most helpful.
>
> [1] Poesia, G., Polozov, O., Le, V., Tiwari, A., Soares, G., Meek, C., & Gulwani, S. Synchromesh: Reliable code generation from pre-trained language models. ICLR 2022

---

> > ### Comment · Reviewer_GDuS · 2023-11-21
> >
> > What I still don't understand is: after the ]], LogicGuide calls Peano, which creates an output (list of one-step inferences). The LLM just continues its generation;  the only control you have over it is to bias the tokens being generated. Presumably the one step inferences are statements in logic and are not tokens that are being biased. How do you get from these statements to a way to bias the logits; presumably there are many ways the LLM could generate the one step conclusions. Or do you actually force the LLM to generate exactly the text of one of  the one-step inferences, in which case where does that text come from?  I could imaging you add a large constant to the logits of first word of each of the one step conclusions and then to subsequent words for the matching conclusions. In this way you are forcing the LLM to state exactly one of the one-step conclusions as its next few  tokens? (Most of your description in the paper seems to be about how to get the input to LogicGuide/Peano, which I'm guessing is the challenging part to implement.) This seems to be very sensitive to which tokens the LLM will generate (e.g., it can't do greedy decoding), but has to fairly sample from its predicted distribution of the next token.
> >
> > When you say "The model then starts to write.." I guess you mean the LLM writes (because you could also output the results of Peano).

---

> > > ### Author Response · Authors · 2023-11-21
> > > **Clarification**
> > >
> > > Thank you for the follow up question. We believe we understand the confusion better now. Let us know if this clarifies!
> > >
> > > In summary, LogicGuide and Peano only influence the LLM's output within `[[` and `]]`. Outside, the model is unconstrained and there are no logit biases. When the model outputs `[[infer:`, this is when we call Peano, which outputs a list of one-step inferences, as you correctly mentioned. Now, we use logit biases to constrain what follows `[[infer:` to be one of the expressions that Peano listed. This might, for example, be completed to ``[[infer:(blue sam)]]``, using after sampling using the model's log-probabilities constrained to the support of the formal inferences that Peano deems valid. Translating this list of strings into which tokens from the model's vocabulary to allow at each step is challenging, and that's where we leverage CSD. Now, after the bracket, the model returns to unconstrained generation. Thus, we do not control how it will interpret the sentence in natural language. However, since the few-shot examples demonstrate that the formal inferences are followed by natural language interpretations of them, the LLM will typically output a sentence that re-interprets the formal term in natural language (so here it migth say ``[[infer:(blue sam)]] Sam is blue.``.
> > >
> > > Although we do not control the "Sam is blue" part, since it is conditioned on the formal inference that comes before, we noticed it to be highly reliable. For the LLM to hallucinate an inference (e.g., if it "wants" to conclude that "Sam is blue" even if that does not logically follow), to follow the demonstrations, it will first try to make an incorrect formal inference right before the natural language description. But the formal inference will not be allowed if it is not correct, so the output will be forced to change course at that point.
> > >
> > > We also note that we apply syntactic constraints to the model's output in the other LogicGuide actions. For example, when the model outputs ``[[axiom:``, we use the logit biases to enforce what comes next to be a valid formal expression. For this, we use the method introduced in CSD to enforce a simple grammar of Peano terms. We ran an additional analysis removing these constraints, and found that 11.5% of GPT-3.5 Turbo's unconstrained formalizations fail to parse or be accepted by Peano. With constraints, we eliminate all of these errors by construction. Although subtler semantic errors can still persist, we can guarantee that we obtain _some_ valid formalization to later constrain the inferences.
> > >
> > > We hope this clarifies, but please let us know if there are further questions!

---

> > > > ### Comment · Reviewer_GDuS · 2023-11-22
> > > >
> > > > In your comment, I am assuming that "the model" is the LLM. If so, it makes sense.
> > > >
> > > > Now I just don't understand how you get the LLM to output [[infer:  or even [[ in the first place. How does it know that it should ask for advice.

---

> > > > > ### Author Response · Authors · 2023-11-22
> > > > >
> > > > > Yes, that's right, the model is the LLM in this case.
> > > > >
> > > > > We get it to output the guided blocks through few-shot prompting, since we include those in the reasoning demonstrated in prompt examples.
> > > > >
> > > > > We hope that clarifies!

---

> > > > > > ### Comment · Reviewer_GDuS · 2023-11-23
> > > > > >
> > > > > > Thanks. That's what I had guessed (but I'm finding my guesses are often wrong ;^)

---

### Official Review · Reviewer_cFZp · 2023-10-30

**Soundness:** 3 good
**Presentation:** 3 good
**Contribution:** 2 fair
**Rating:** 5
**Confidence:** 4

**Summary:**

This paper studies logical reasoning in natural language with LLMs. Whereas a number of existing approaches may arrive at the correct answer with a wrong reasoning chain, this work proposes an approach to guide the LLM generations using a logic solver that constrains the space of possible generations to those that are logically valid. With this approach, while there can still be errors in the translation stage (i.e. the stage where the LLM translates from natural language to logical form), the logical conclusions made on those translations are valid. Experimental results are shown on multiple datasets including ProofWriter, PrOntoQA, Syllogism Validity, LegalBench, and ReClor.

**Strengths:**

- Logical reasoning (or more generally multi-hop reaosning) in natural language with LLMs is an important area of research.
- Showing results both for prompting and finetuning.
- The writing was mostly clear and easy to follow.
- The reported improvements for ReClor could be quite encouraging.

**Weaknesses:**

- Most of the experiments are done on the ProofWriter and the PrOntoQA datasets. Both these datasets have been constructed by turning logical theories into natural language using very simple templates. This is especially true for the PrOntoQA dataset where each sentence is of the format "X is Y" which is simply equivalent to (X, is, Y) in the triple notation. For this reason, while these datasets are appropriate benchmarks for measuring the general reasoning capacity of off-the-shelve LLMs, I do not think they are good benchmarks for the model proposed in this paper (translating these datasets back into their logical form is just too easy for nowadays LLMs). For this reason, while those results could be good sanity checks, I don't think they truly represent the merit of the proposed approach. They highly overestimate the performance we can expect on real tasks but highly underestimating how difficult it is to translate an actual natural language passage into logical form.
- The failure example highlighted in Page 6 (translating to (sees A B) in one place and (see A B) in another) makes me worry about the applicability of the proposed approach to reasoning problems beyond synthetic tasks such as ProofWriter and PrOntoQA. It also makes me  think that BoardgameQA might have been a slightly better dataset to use. While it has also been generated synthetically by converting logical theories into textual format, the missing knowledge piece of it makes it better resemble real-world problems, and makes for a good test to see the extent of the "see" vs" sees" problem in the proposed approach.
- While the results on the ReClor dataset are quite encouraging, I find them quite surprising as well for multiple reasons. 1- Given that the model is finetuned only on 120 samples, and considering the size of the models used, I would expect that the models should just overfit to those examples without any task transfer. 2- If I understand correctly, the finetuning is not on a mixture of the original data and the 120 data points, so I would expect that the model's general task solving ability should go down. 3- The ProofWriter and PrOntoQA datasets only require deductively applying the modus ponens rule, whereas the ReClor dataset requires more complicated rules and reasoning. For these reasons, I found the improvements a bit surprising and the provided explanation does not give much insights.

**Questions:**

- On which categories from Table 2 of the BoardgameQA paper do you expect your approach to fail/succeed? And why?
- Given that the results in Table 6 are tested in a zero-shot setting, how do you extract the final answer? Is it possible that after finetuning on the 120 examples, the model mainly just learns to produce outputs in the specified format making it easier to extract the final answer (and hence higher predictive accuracy)?

---

> ### Author Response · Authors · 2023-11-15
> **Author response**
>
> Thank you for the thoughtful comments on our work! We address the questions below, and are happy to engage further.
>
> > While the results on the ReClor dataset are quite encouraging, I find them quite surprising as well for multiple reasons. 1- Given that the model is finetuned only on 120 samples, and considering the size of the models used, I would expect that the models should just overfit to those examples without any task transfer. 2- If I understand correctly, the finetuning is not on a mixture of the original data and the 120 data points, so I would expect that the model's general task solving ability should go down. 3- The ProofWriter and PrOntoQA datasets only require deductively applying the modus ponens rule, whereas the ReClor dataset requires more complicated rules and reasoning. For these reasons, I found the improvements a bit surprising and the provided explanation does not give much insights.
>
> Thank you for the question -- we will update the paper with a better discussion of the transfer results, which is indeed needed. Although we have few public technical details on the GPT-3.5 fine-tuning internals, the fine-tuning method is most likely a low-rank adaptation (like LoRA [1], which we used for our experiments on LLama), so the parameters of the base model are likely unchanged. Instead, a small number of parameters is learned on top of the model to adjust its behavior to the new examples. In general, parameter-efficient fine-tuning methods like LoRA tend to not overfit as easily or degenerate the model's behavior because they operate with relatively low capacity. This is also why we don't need many examples for fine-tuning (the OpenAI documentation recommends starting with an even smaller number, like 50, and iterating from there, which corroborates this). Our understanding is that low-rank fine-tuning is not necessarily "teaching the model completely new skills" (such as inference patterns unseen during pre-training), but rather reinforcing skills that were already present in the training data, at the expense of less relevant ones for the task. In turn, fitting higher-quality examples yields better results.
>
> > On which categories from Table 2 of the BoardgameQA paper do you expect your approach to fail/succeed? And why?
>
> LogicGuide can in principle be applied to all of those types of rules and comparisons, except Ortography (since Peano does not currently have a representation for strings as sequences of characters). For problems where there are missing assumptions, we can ask the model to formally state those for use in inference. We previously ran a preliminary experiment with this using PrOntoQA False (where the rules violate world knowledge), where instead of giving the model the premises and asking it to reason with them, we gave it the reasoning trace, without any premises, and asked it to first formalize assumptions that would enable the argument to work, and only then proceed with the formal argument. We found few-shot prompting for completing with the assumptions to be effective, and we believe a version of this could also work for the kinds of gaps that happen in BoardgameQA, where the model also needs to identify and complete missing assumptions in the problem (e.g., that "every dog is a mammal" to connect "sam is a dog" and "every mammal …"). We are running a scaled up version of this experiment and will update the paper with results on it soon (and let you know here as well).
>
> > Given that the results in Table 6 are tested in a zero-shot setting, how do you extract the final answer? Is it possible that after finetuning on the 120 examples, the model mainly just learns to produce outputs in the specified format making it easier to extract the final answer (and hence higher predictive accuracy)?
>
> In the zero-shot setting, we prompted GPT-3.5 Turbo with an explanation of both the task and the desired output format, and we did allow for some deviation by looking at the outputs of the base model and matching extra patterns. We observed that the original model before our fine tuning did follow the desired format, and only deviated from it in a few predictable ways (e.g., instead of saying just "Answer: A" as we asked, it would often say "Answer: A - <<option text here>>"). We note that the fine-tuned model was still subject to these variations because the format of ReClor was not exactly the one during training (e.g., here there were several options for the answers, while the training examples were just true/false).
>
> Thank you again for the attentive review! Please let us know if we can clarify further. We'll update you here once we have the results on the experiments with missing assumptions.
>
> [1] HU, Edward J. et al. LoRA: Low-Rank Adaptation of Large Language Models. ICLR 2022

---

> > ### Author Response · Authors · 2023-11-21
> > **Author response**
> >
> > We thank the reviewer again for the comments! We'd like to follow up on the experiment with incomplete premises.
> >
> > We ran our method on an additional task that demonstrates the potential to work with incomplete premises. We took PrOntoQA-False, and designed a task where instead of being given facts, rules and a question, the model is given an argument (a sequence of conclusions), and has to fill in the assumptions that make the argument work. We used PrOntoQA-False since it leverages counter-intuitive assumptions, which we found to be harder for the model. Using LogicGuide, we had the model first output formal assumptions _that are only implicit_ in the argument (using `[[axiom:]]`), and then finally formalize the argument itself. We considered success when the model was able to formally obtain the conclusion of the original argument.
> >
> > GPT 3.5-Turbo with LogicGuide was able to do this task in 87% of the cases, with an example case given below:
> >
> > ```
> > Informal argument (given): Wren is an animal. Wren is a feline. Wren is a sheep. Wren is a mammal. Wren is a vertebrate. Wren is luminous.
> > (model):
> > Implicit premises: All animals are felines [[axiom:(animal 'x) -> (feline 'x)]]. Felines are sheep [[axiom:(feline 'x) -> (sheep 'x)]]. Sheep are mammals [[axiom:(sheep 'x) -> (mammal 'x)]]. Mammals are vertebrates [[axiom:(mammal 'x) -> (vertebrate 'x)]]. Vertebrates are luminous [[axiom:(vertebrate 'x) -> (luminous 'x)]].
> > Formalized argument: Wren is an animal [[axiom:(animal wren)]]. Wren is a feline [[infer:(feline wren)]]. Wren is a sheep [[infer:(sheep wren)]]. Wren is a mammal [[infer:(mammal wren)]]. Wren is a vertebrate [[infer:(vertebrate wren)]]. Wren is luminous [[infer:(luminous wren)]].
> > ```
> >
> > In the failure cases, in around half the model tried to output a more intuitive assumption that was not strong enough (e.g. saying "every number is either odd or even", whereas the argument was implying that every number is even), or simply missing some of the premises, thus being unable to ultimately infer the conclusion.
> >
> > We only tried once for each problem, but it is possible to explore combining this idea with a more structured approach (e.g., Tree of Thoughts) that can explore multiple possible implicit assumptions. In turn, LogicGuide can serve as a source of intermediate feedback on whether the extra assumptions license the desired conclusions as intended.
> >
> > We're happy to clarify further. We'd also like to ask if your questions about the transfer experiments have been addressed, as we're happy to provide more detail before the end of the rebuttal period.

---

### Official Review · Reviewer_HHEy · 2023-11-01

**Soundness:** 3 good
**Presentation:** 2 fair
**Contribution:** 2 fair
**Rating:** 5
**Confidence:** 3

**Summary:**

This paper introduces a novel tool, termed "guide," designed to ensure that language models engage in sound step-by-step reasoning. As a primary illustration, LogicGuide utilizes general logical reasoning systems to guide models towards producing logically consistent explanations. Experimental results indicate that LogicGuide enhances the performance of language models, in terms of reasoning accuracy, reducing content effects, self-learning and generalization.

**Strengths:**

1. The paper introduces a novel logical guidance framework designed to aid LLMs in performing logical inference. The method employs the most general form of deductive reasoning, making it versatile across a range of reasoning scenarios.
2. Experiments across multiple datasets validate that LogicGuide enhances the performance of language models. The paper also provides specific examples demonstrating its efficacy in mitigating the impact of unwarranted prior assumptions and performing self-learning.

**Weaknesses:**

1. The proposed method necessitates a reliance on a complex formalization process during training and inference.

2. The scenarios considered in the paper seem a bit limited. Despite experimenting on diverse datasets, the nature of problems within them appear quite similar. In more generalized contexts, it might be challenging to formalize and identify corresponding actions, such as `objects`, `relations`, etc.

3. The paper's primary contribution, namely, how to harness logic to ensure output consistency, seems to overlap with prior work on the Peano theorem and the constrained Semantic Decoding algorithm, which weakens the novelty of the current research.

4. It seems the proposed idea is similar to the idea in Logic-LM. The authors did not discuss their differences.

   Logic-LM: Empowering Large Language Models with Symbolic Solvers for Faithful Logical Reasoning https://arxiv.org/abs/2305.12295

**Questions:**

1. How likely that encountering a formalization failure may happen, and are there strategies in place to minimize formalization errors?
2. To what extent does using constrained generation reduce the reasoning space, so as to mitigate the issue of "logical inferences made next can have a potentially large set of answers"? Is it possible that still there may be a considerably large set of answers, if so, how does your method decide on the the most appropriate content to generate next?
3. Discussions on generalization involve models bootstrapped from other formalizable tasks. In scenarios challenging to formalize, what amount of preparatory work, such as the number of samples of formalizable tasks, is essential to ensure the model with strengthened generalization inference capabilities? If in the absence of abundant corresponding simpler tasks, how to generalize "guide" in broader scenarios?

---

> ### Author Response · Authors · 2023-11-15
> **Author response**
>
> We thank the reviewer for the thoughtful comments, and the encouraging remark that the approach is "versatile across a range of reasoning scenarios"! We respond to the questions and concerns below, and are happy to discuss these further.
>
> > The proposed method necessitates a reliance on a complex formalization process during training and inference.
>
> While this is true for problems that can be adequately formalized, our results (Section 4.4) show that bootstrapping GPT-3.5 Turbo on such problems, where formalization helps during training, can also improve reasoning when not formalizing at inference time.
>
> > In more generalized contexts, it might be challenging to formalize and identify corresponding actions, such as objects, relations, etc.
>
> We agree that this can be hard in more naturalistic problems. Our experiments in Section 4.4 show that bootstrapping on simpler problems can still help even when not formalizing at inference time.
>
> > Overlap with prior work on the Peano theorem and the constrained Semantic Decoding algorithm
>
> Our framework of guides is a novel interface between these components. Peano itself is simply a formal theorem proving environment, with no natural language component. Moreover, while Constrained Semantic Decoding was originally used in cases where the entire output is constrained, our framework of guides generalizes this to allow the model to selectively turn the guide tool (and therefore the constraints) on and off during generation. Thus, we build on these prior works but ultimately implement new capabilities.
>
> > It seems the proposed idea is similar to the idea in Logic-LM.
>
> Thank you for the reference. We'll add it to our Related Work section. In summary, works like Logic-LM use the LLM to parse the problem, and then offload reasoning entirely to an external engine. Thus, the model does not produce a reasoning trace in natural language (a "chain-of-thought"), since reasoning is carried out by the external tool (note that none of the tools used in Logic-LM produce such a trace either). In our work, Peano is only used to limit what inferences the model itself can do, but ultimately the LLM is still the reasoner, and still produces a chain-of-thought. We need these reasoning traces to fine-tune the model itself in a self-improvement fashion (which we show in Sections 4.3 and 4.4).
>
> > How likely that encountering a formalization failure may happen, and are there strategies in place to minimize formalization errors?
>
> It varies from model to model - the OpenAI models were fairly good at formalization in our experiments, whereas LLaMA lagged behind. For the formalization blocks, LogicGuide still imposes constraints on well-formedness during generation to minimize errors. Namely, the constraints guarantee that the model will output expressions that are syntactically valid and coherent with its own proposition and object annotations, thus avoiding errors due to such inconsistencies. These constraints were triggered in around 5-10% of the cases depending on the dataset for text-davinci-003 and gpt-3.5-turbo, where otherwise there would have been formalization errors. We are running a more in-depth analysis on this and will add it to the paper soon.
>
> > To what extent does using constrained generation reduce the reasoning space, so as to mitigate the issue of "logical inferences made next can have a potentially large set of answers"? Is it possible that still there may be a considerably large set of answers, if so, how does your method decide on the the most appropriate content to generate next?
>
> The number of possible valid inferences for the datasets we used were in the order of dozens in the worst case. In these cases, the language model is constrained to generate from that set when it opens an [[infer]] block, and uses its own log-probabilities to decide what is most appropriate in the current context / solution. Thus, we allow the model to choose its logical inferences while constraining it to logically sound ones. Allowing the language model to decide amongst logically valid continuations is unique to our guide approach.
>
> > In scenarios challenging to formalize, what amount of preparatory work, such as the number of samples of formalizable tasks, is essential to ensure the model with strengthened generalization inference capabilities?
>
> This is a good question. In general, we cannot strictly guarantee that positive transfer will happen between the training and inference tasks. However, using low-rank fine-tuning methods [1], we observe that even a few hundred high-quality training examples can improve reasoning performance. In turn, the guided models are able to produce higher quality training examples for this purpose, as our results show.
>
> Please let us know if we have clarified your concerns. We're happy to discuss these further or provide more information!
>
> [1] HU, Edward J. et al. LoRA: Low-Rank Adaptation of Large Language Models. ICLR 2022

---

> > ### Author Response · Authors · 2023-11-21
> > **Aid in formalization errors**
> >
> > We thank the reviewer again for the valuable comments. We'd like to report an updated analysis on formalization errors and how our approach prevents them.
> >
> > As we mention in the paper (Section 3.2), when activated by the model during formalization, LogicGuide enforces syntactic constraints to ensure that the model outputs well formed expressions. Thus, we are always able to extract some formalization from the model's output *by construction*. We ran an additional experiment where we use the LogicGuide prompt but don't apply such constraints, instead letting the model output the formalization on its own. Note that this approach is similar to LogicLM, in that the model is prompted to translate the problem into a formal form without any interference in its formalization.
> >
> > Without our constraints during generation, out of 400 problems (in a mixture of ProofWriter and PrOntoQA), GPT-3.5 Turbo produced malformed formalizations in 46 cases (*11.5%* of the time). In these cases, the formalization is unusable, due to syntax errors or inconsistent arities in predicates. One example from ProofWriter is shown below:
> >
> > ```
> > (...) If something is [[prop:rough]] and [[prop:furry]] then it is [[prop:young]]. [[axiom:((rough 'x) -> (furry 'x)) -> (young 'x))]]
> > ```
> >
> > There are two subtle syntax errors here: (a) the first two premises are enclosed in parentheses where in Peano they shouldn't, and (b) there's an extra closing parentheses at the end. Since we leverage a simple grammar of first-order expressions to apply constraints with LogicGuide, we completely avoid these errors by construction, always sampling a formula that we're able to parse. Thus, with these constraints, we obtain 100% of valid formalizations. Semantic errors can still persist, since we cannot formally guarantee that the model has correctly interpreted the input sentences, even if it produces a valid formal interpretation. However, this step significantly improves our approach. We'd like to also notice that applying constraints to aid in formalization is unique to our method, since prior work (like LogicLM or Faithful-CoT) work in a two-phase manner, where first the LLM fully formalizes the input, and then the solver is called on the result. In this case, however, if the LLM outputs an invalid formalization, the solver cannot do anything to its input.
> >
> > Since the rebuttal period is ending soon, we'd like to check if we have properly addressed your questions. Please let us know if we can clarify anything further!

---

> > > ### Comment · Reviewer_HHEy · 2023-11-23
> > >
> > > I have read the author responses.

---

### Meta-Review · Area_Chair_JGif · 2023-12-03

**Metareview:**

The paper addresses an important problem, namely how to realise certified deductive reasoning with language models. To this end, it shows how to use a general system for logical reasoning as guide tool for the language model. While the reviewers have mixed opinion, they all agree that the direction taken is important. Two of them present salient arguments about the suitability of this paper for ICLR in its current form such as there are still failures such as (sees A B) versus (see A B). As the positive reviewer puts it "the high level idea seems good (but the details I'm not so clear about). The results are very good." In my opinion, the paper will benefit from another cycle to provide all details of the method and also clarify the translation experiment, in particular since the main idea of  constraints generation has been presented elsewhere.

**Justification For Why Not Higher Score:**

Indeed, an important problem, but the method heavily relies on a constraints generation technique already presented elsewhere.

**Justification For Why Not Lower Score:**

N/A

---

### Decision · Program_Chairs · 2024-01-16

Reject